# ERGO: EFFICIENT HIGH-RESOLUTION VISUAL UNDERSTANDING FOR VISION-LANGUAGE MODELS

**Jewon Lee**[1][*]   **Wooksu Shin**[1][*]   **Seungmin Yang**[1]   **Ki-Ung Song**[1]   **DongUk Lim**[1]
**Jaeyeon Kim**[1]   **Tae-Ho Kim**[1][†]   **Bo-Kyeong Kim**[1][†]

[1]**Nota Inc.**
`{jewon.lee, wooksu.shin, thkim, bokyeong.kim}@nota.ai`

## ABSTRACT

Efficient processing of high-resolution images is crucial for real-world vision–language applications. However, existing Large Vision-Language Models (LVLMs) incur substantial computational overhead due to the large number of vision tokens. With the advent of "thinking with images" models, reasoning now extends beyond text to the visual domain. This capability motivates our two-stage "coarse-to-fine" reasoning pipeline: first, a downsampled image is analyzed to identify task-relevant regions; then, only these regions are cropped at full resolution and processed in a subsequent reasoning stage. This approach reduces computational cost while preserving fine-grained visual details where necessary. A major challenge lies in inferring which regions are truly relevant to a given query. Recent related methods often fail in the first stage after input-image downsampling, due to *perception-driven reasoning*, where clear visual information is required for effective reasoning. To address this issue, we propose **ERGO** (Efficient Reasoning & Guided Observation) that performs *reasoning-driven perception*—leveraging multimodal context to determine where to focus. Our model can account for perceptual uncertainty, expanding the cropped region to cover visually ambiguous areas for answering questions. To this end, we develop simple yet effective reward components in a reinforcement learning framework for coarse-to-fine perception. Across multiple datasets, our approach delivers higher accuracy than the original model and competitive methods, with greater efficiency. For instance, ERGO surpasses Qwen2.5-VL-7B on the V* benchmark by **4.7** points while using only **23%** of the vision tokens, achieving a **3×** inference speedup. The code is available at `https://github.com/nota-github/ERGO`.

## 1 INTRODUCTION

High-resolution image processing is crucial to achieve strong performance in real-world applications with large vision–language models (LVLMs) (Liu et al., 2024a; Wang et al., 2024a; Vasu et al., 2025). Recent reinforcement learning (RL)-based post-training approaches (Wang et al., 2025a; Zheng et al., 2025b) have explored the idea of "thinking with images" (OpenAI, 2025), enabling LVLMs to reason not only through text, but also within the visual modality itself. By reasoning over cropped image features with bounding-box coordinates, these models can attend to local high-fidelity objects and capture fine-grained details, leading to significant improvements in high-resolution benchmarks.

Despite these advances, processing high-resolution input remains a major challenge. LVLMs must handle a massive number of vision tokens, resulting in prohibitive computational costs. A straightforward solution (Zhou et al., 2025; Yang et al., 2025), is to reduce the input resolution, which results in fewer vision tokens but inevitably discards fine-grained details critical to reasoning. The two-stage "coarse-to-fine" pipeline embodies this principle: it first queries the model with a coarse-grained image for initial reasoning over task-relevant regions; and then selectively localizes and re-encodes sub-images at higher resolution with finer granularity for subsequent reasoning. Crucially, discovering relevant regions from downsampled image input is fundamental to overall performance, as it guides the model to focus its capacity on informative areas.

Fig. 1 illustrates this challenge and our solution. In Fig. 1(a), DeepEyes (Zheng et al., 2025b) performs well when the target object remains clearly visible (i.e., correctly identifying a straw in the

---

[*]These authors contributed equally and are listed alphabetically.
[†]Corresponding authors.

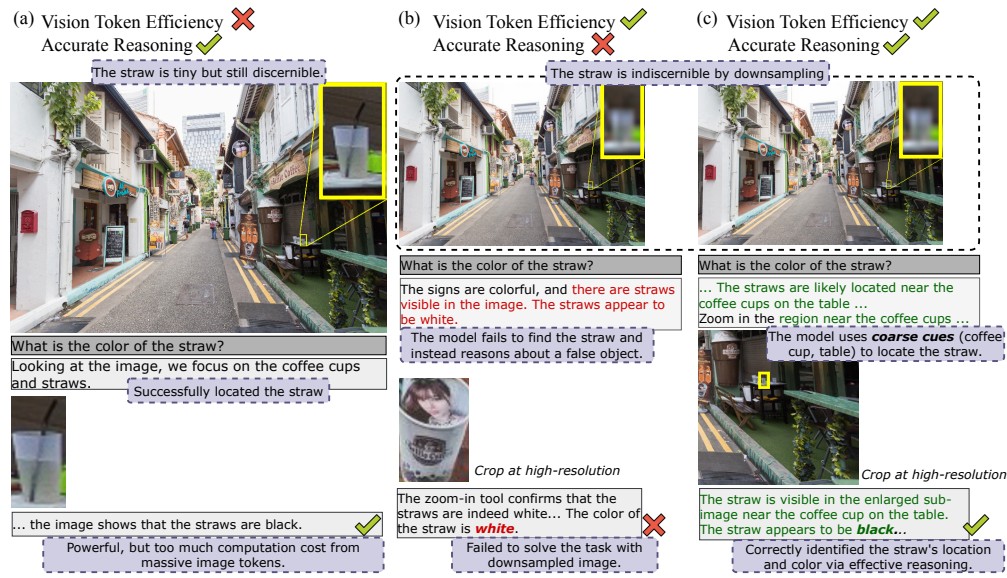

Figure 1: **Comparison with prior work on high-resolution visual reasoning.** The yellow box marks the target object, which becomes indiscernible after input-image downsampling. (a) Deep-Eyes (Zheng et al., 2025b) succeeds when the object remains discernible, but at the cost of a large number of vision tokens. (b) DeepEyes (Zheng et al., 2025b) fails when the object is indiscernible at low resolution, where fewer vision tokens are available. (c) Our ERGO performs reasoning-driven perception, correctly answering the question even on low-resolution images.

high-resolution image), but it requires processing a large number of vision tokens. Relevant models (Wang et al., 2025a; Su et al., 2025a; Zheng et al., 2025b) are typically designed in this *perception-driven reasoning* paradigm, where the model first localizes a tightly bounded target and then reasons over it. As a result, their training tends to overlook downsampled visual inputs. While effective at full resolution, this paradigm becomes a bottleneck in efficiency-oriented scenarios.

After input-image downsampling for a smaller number of vision tokens (see Fig. 1(b)), the straw becomes indistinguishable, causing DeepEyes (Zheng et al., 2025b) to miss it and incorrectly focus on more discernible objects. In contrast, under such pixel-constrained conditions, our approach (Fig. 1(c)) highlights that *reasoning-driven perception* (i.e., including contextually inferable regions such as straws near coffee cups on tables) is far more beneficial, since selecting the correct region enables recovery of the original resolution in that area.

We introduce **ERGO** (Efficient Reasoning & Guided Observation), whose training objective is explicitly aligned with vision-processing efficiency in a reinforcement learning (RL) framework. It rewards the inclusion of all task-relevant regions, while implicitly incentivizing the incorporation of auxiliary context. This design enables the model to handle ambiguity without being restricted to precise localization, learning that exact identification of individual objects is not always optimal and that reasoning with contextual knowledge is often more beneficial. By aligning visual exploration with efficiency objectives, our approach enables LVLMs to achieve improved efficiency without sacrificing fine-grained reasoning ability. Our key contributions can be summarized as follows.

○ **Efficient coarse-to-fine pipeline.** We introduce a two-stage reasoning pipeline that first processes low-resolution inputs to identify task-relevant regions and then re-encodes them at higher resolution. The pipeline reduces computational cost while preserving essential information.

○ **Reward for reasoning-driven perception.** With our proposed reward, the policy model learns that relying solely on accurate object localization is *not* always optimal and that contextual knowledge can often be *more* beneficial. To our knowledge, we are the first to demonstrate the significance of this insight for high-resolution visual processing in LVLMs.

○ **State-of-the-art performance with fewer vision tokens.** ERGO surpasses competitive methods (Huang et al., 2025b; Yang et al., 2025; Zheng et al., 2025b; Wang et al., 2025a; Su et al., 2025a; Lai et al., 2025) in accuracy on multiple high-resolution benchmarks, while reducing vision token counts and delivering practical speedups.

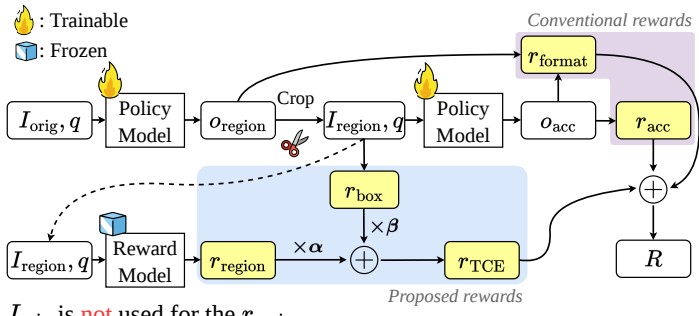

$I_{\text{orig}}$ is not used for the $r_{\text{region}}$.

Figure 2: **Overview of RL-based training pipeline.** The blue background highlights the components of the proposed TCE reward. The purple background highlights the conventional rewards adopted by most reasoning LVLMs.

## 2 MOTIVATION

We examine whether appending a critical high-resolution sub-image to a low-resolution image input can enhance model performance. For the experiments, we used Qwen2.5-VL (Bai et al., 2025) with *pixel constraints* to resize input images. Specifically, a setting of $N{\times}28{\times}28$ in its image processor caps the maximum number of vision tokens at $N$. We varied the input resolution by controlling $N$ and included the ground-truth (GT) original-resolution sub-image as an auxiliary input. Tab. 1 shows that using the GT full-fidelity sub-image does not degrade performance, even when the model has not been explicitly trained under such conditions. This finding indicates that high-resolution access to task-relevant regions is sufficient, whereas redundant tokens merely reduce efficiency.

| Pixel const. | Task-relevant region | V* |
|---|---|---|
| 16384×28×28 | ✗ | 77.0 |
| 1280×28×28 | ✗ | 64.9 |
| 640×28×28 | ✗ | 56.5 |
| 640×28×28 | ✓ | **77.0** |

Table 1: **Effectiveness of high-resolution task-relevant cues.** "Task-relevant region" denotes whether the annotated GT sub-image at original resolution is appended to the input. Evaluation was conducted using Qwen2.5-VL-7B on the V* benchmark.

Now that we have shown the effectiveness of task-relevant regions, the next question is whether existing models can autonomously identify such regions. A straightforward strategy might integrate a powerful "thinking with images" model (Zheng et al., 2025b; Su et al., 2025a; Huang et al., 2025b; Wang et al., 2025a) into the coarse-to-fine pipeline, predicting grounded coordinates and cropping the corresponding high-resolution sub-images. However, our results show that existing RL-trained reasoning models struggle to perform this task under low-resolution inputs (see Tab. 2). This highlights the need for approaches that can robustly identify informative regions even when coarse visual cues are the only available signals, rather than relying solely on clearly discernible objects.

## 3 PROPOSED METHOD

Our objective is to develop remarkable *reasoning-driven perception* models that can reason over where to focus. Fig. 2 presents our RL-based training pipeline, whose forward process is as follows:

○ Given a pair of original image $I_{\text{orig}}$ and text query $q$, the policy model $\pi_\theta$ produces output $o_{\text{region}} \sim \pi_\theta(\cdot \mid I_{\text{orig}}, q)$, which includes candidate bounding-box coordinates (indicating the region relevant to the query) and a thinking trace.

○ Next, the image $I_{\text{region}}$ corresponding to the bounding box is cropped from the original image $I_{\text{orig}}$ to feed into the reward model: $I_{\text{region}} \leftarrow \text{crop}(I_{\text{orig}}, o_{\text{region}})$.

○ Then, the policy $\pi_\theta$ generates an answer $o_{\text{acc}} \sim \pi_\theta(\cdot \mid [I_{\text{region}}, q], [I_{\text{orig}}, o_{\text{region}}])$ in a multi-turn conditioned setting, based on both the past interaction (i.e., original image $I_{\text{orig}}$ and predicted bounding box $o_{\text{region}}$) and the current query pair (i.e., cropped region $I_{\text{region}}$ and text query $q$).

The strength of ERGO lies in well-designed reward components for coarse-to-fine vision-grounded reasoning, detailed as follows.

## 3.1 REWARD DESIGN

### 3.1.1 PROPOSED REWARD

**Region-verification reward.** In many thinking-with-images studies (Huang et al., 2025b; Su et al., 2025a; Zheng et al., 2025b), a reward model $\mathcal{R}$ takes the original image together with the cropped region and query, producing its output $o_{RM} \sim \mathcal{R}(\cdot|I_{orig}, I_{region}, q)$ to guide the policy model. However, we argue that *feeding the original image $I_{orig}$ into the reward model* is *sub-optimal*: the model may rely on the original image instead of the cropped region, introducing unnecessary hints to the query and thereby weakening the objective of self-contained cropping (i.e., ensuring the cropped region alone provides sufficient cues). This issue is particularly problematic for coarse-to-fine visual grounded reasoning, which we adopt for efficiency, because low-resolution input images contain little evidence (as target objects are often indiscernible), making self-contained crops essential for question answering.

To address this issue, we propose the region-verification reward $r_{region}$, where task performance is evaluated using only the cropped region and the query, *without* access to the original image. We reframe the complex task of locating the optimal region into the simpler task of answering the question with a single cropped image. If the reward model's prediction matches the GT answer $o_{GT}$, the policy model receives a reward:

$$o_{RM} \sim \mathcal{R}(\cdot|I_{region}, q), \quad r_{region} = \mathbb{1}\left[\text{match}(o_{RM}, o_{GT})\right]. \tag{1}$$

This design encourages the policy model to identify informative, task-relevant regions that preserve sufficient information for accurate reasoning, without the need for additional annotations. In practice, we use a frozen reward model, Qwen2.5-VL-72B-Instruct (Bai et al., 2025).

**Box adjustment reward.** Although the region reward effectively encourages task-guided cropping, a key challenge emerges during early training: the policy model may exploit a trivial strategy by consistently selecting the entire image. While this would be a reasonable shortcut for maximum region reward, since the whole image is necessarily self-contained for the task, it limits efficient inference due to excessive token costs from processing the full-resolution image.

To mitigate this issue, we introduce a complementary reward signal that regularizes the size of the selected region. Specifically, the box adjustment reward $r_{box}$ is computed with a step function that penalizes overly large crops based on the area ratio of the selected region to the original image; it effectively prevents the model from consistently grounding the entire image:

$$r_{box} = \mathbb{1}\left[\frac{\text{Area}(I_{region})}{\text{Area}(I_{orig})} \leq \gamma\right]. \tag{2}$$

Determining an ideal value of $\gamma$ is crucial for our approach: low enough to prevent degenerate solutions (e.g., selecting the full image as the crop) during training, yet high enough to allow flexibility in region selection. To this end, we examined the training split of popular LVLM reasoning-related datasets with answer-aligned bounding box annotations (e.g., TreeVGR (Wang et al., 2025a), VisCoT (Shao et al., 2024a), V* (Wu & Xie, 2023), VGR (Wang et al., 2025b)). Fig. 3 shows that most GT regions relevant to question answering occupy less than 60% of the full image. Based on this analysis, we set $\gamma = 0.6$ for efficient and effective bounding box adjustment.

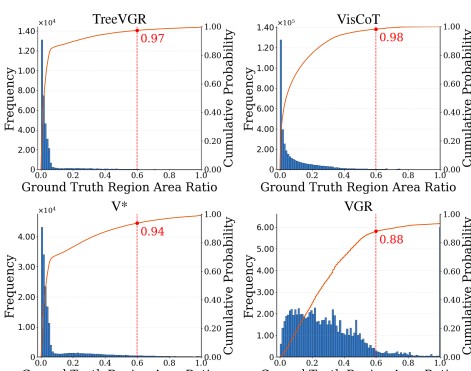

Figure 3: **Analysis of query-relevant GT regions in training data.** Most GT regions span less than 60% of the full image area.

**Task-driven contextual exploration (TCE) reward.** Based on the collaborative nature of the region reward and box adjustment reward, we combine them to form our main reward $r_{TCE}$:

$$r_{TCE} = \alpha \cdot r_{region} + \beta \cdot r_{box}. \tag{3}$$

Here, $\alpha$ and $\beta$ are weighting coefficients, set to $\alpha = 1$ and $\beta = 0.5$. This enables the policy model to learn robust and efficient region selection strategies for vision-grounded reasoning.

### 3.1.2 CONVENTIONAL REWARD

**Accuracy reward.** The TCE reward is effective to guiding the policy model to select task-relevant regions. However, it only indirectly promotes correct question-answering, creating a potential mismatch between the training objective and the final evaluation. To bridge this gap, we use an accuracy reward (DeepSeek-AI et al., 2025), which is assigned when the policy model's output $o_{acc}$ matches the GT answer: $r_{acc} = \mathbb{1}[\mathrm{match}(o_{acc}, o_{GT})]$. This component complements the TCE reward by directly optimizing for question-answering accuracy.

**Format reward.** This reward enforces the adhesion to a predefined output structure using special tags (DeepSeek-AI et al., 2025). A reward is given if the reasoning is correctly enclosed within `<think></think>` tags, the final answer within `<answer></answer>` tags, and a `<zoom></zoom>` tag is included when region selection is performed: $r_{format} = \mathbb{1}[o_{region}, o_{acc}$ follow expected format $]$. This mechanism encourages the model to maintain well-formed outputs that can be reliably parsed and evaluated throughout training and inference.

### 3.1.3 FINAL REWARD FORMULATION

The overall reward function is defined as a linear combination of three components (i.e., the TCE reward, the accuracy reward, and the format reward):

$$R = r_{TCE} + r_{acc} + r_{format}. \tag{4}$$

### 3.2 LEARNING ALGORITHM

We adopt Grouped Reward Policy Optimization (GRPO) (Shao et al., 2024b) as our RL framework, leveraging its sample-efficient optimization in grouped feedback settings (see the pseudo-code in Sect. A for details). Through this effective RL training, ERGO acquires *reasoning-driven perception* capabilities when presented with low-resolution, target-indiscernible inputs.

## 4 EXPERIMENTAL SETUP

**Training setup.** We use Qwen2.5-VL-7B-Instruct (Bai et al., 2025) as the policy model and Qwen2.5-VL-72B-Instruct (Bai et al., 2025) as the frozen reward model. Our training data consists of a subset of ArxivQA (Li et al., 2024) and the V* training set (Wu & Xie, 2023), following Zheng et al. (2025b). Training was conducted with a global batch size of 128, using 8 rollouts per example on 4 H100 GPUs. See Sect. B for full details.

**Baselines.** We compare our approach against two categories of RL-based post-training methods:

○ *Efficiency-oriented* models share our objective of efficient high-resolution vision–language understanding. MGPO (Huang et al., 2025b) directly leveraged the multi-turn pipeline but with a single reasoning stage. VisionThink (Yang et al., 2025) does not select a sub-region of the original image; instead, it employs a mechanism whereby the model, given a downsampled image, determines whether the full high-resolution image should be processed for the task.

○ *Non-efficiency-oriented* models are considered due to their strong grounding capabilities, which could still benefit the coarse-to-fine pipeline. DeepEyes (Zheng et al., 2025b), PixelReasoner (Su et al., 2025a) and MiniO3 (Lai et al., 2025) are not trained for efficiency, but can be adapted to coarse-to-fine scenarios. Though TreeVGR (Wang et al., 2025a) is inherently incompatible with coarse-to-fine settings, as it performs text-only reasoning over bounding-box coordinates rather than visual re-encoding, we include it as a baseline for its strong grounding performance.

**Benchmarks.** We utilize high-resolution visual question answering (VQA) benchmarks including V* (Wu & Xie, 2023), HR-Bench (Wang et al., 2024b), MME-RWL (Zhang et al., 2024), TreeBench (Wang et al., 2025a) and VisualProbe (Lai et al., 2025), as our objective is efficient high-resolution

| Pixel Const. | Model | V* Bench | HR Bench[4K] | HR Bench[8K] | MME-RW[Lite] | TreeBench | VisualProbe | Average |
|---|---|---|---|---|---|---|---|---|
| 16384×28×28 | Qwen2.5-VL-7B-Inst. | 77.0 | 71.1 | 67.1 | 46.7 | 40.0 | 29.2 | 52.4 |
| 1280×28×28 | Qwen2.5-VL-7B-Inst. | 64.9 | 65.6 | 56.5 | 42.6 | 40.0 | 15.2 | 44.8 |
| | *Non-efficiency-oriented Post Training Methods* | | | | | | | |
| | PixelReasoner (Su et al., 2025a) | 74.5 | 66.9 | 61.3 | 49.8 | 40.4 | 31.8 | 52.1 |
| | DeepEyes (Zheng et al., 2025b) | 78.5 | 66.0 | 60.0 | 48.9 | 40.2 | 39.0 | 52.5 |
| | TreeVGR[‡] (Wang et al., 2025a) | 76.4 | 66.4 | 60.4 | 47.5 | **48.2** | 26.6 | 53.1 |
| | MiniO3 (Lai et al., 2025) | 81.2 | 70.0 | 65.9 | 36.7 | 36.3 | 39.0 | 52.2 |
| | *Efficiency-oriented Post Training Methods* | | | | | | | |
| | MGPO[†] (Huang et al., 2025b) | 77.5 | 69.8 | 61.1 | 44.4 | 39.3 | 33.4 | 52.6 |
| | VisionThink[‡] (Yang et al., 2025) | 73.8 | 66.1 | 65.8 | 49.0 | 41.0 | 17.3 | 49.4 |
| | **ERGO** | **83.8** | **73.0** | **69.9** | **52.6** | 41.7 | **42.7** | **58.4** |
| 640×28×28 | Qwen2.5-VL-7B-Inst. | 56.5 | 57.0 | 49.9 | 40.3 | 40.2 | 10.9 | 40.3 |
| | *Non-efficiency-oriented Post Training Methods* | | | | | | | |
| | PixelReasoner (Su et al., 2025a) | 67.2 | 66.5 | 59.9 | 47.7 | 42.3 | 23.9 | 48.9 |
| | DeepEyes (Zheng et al., 2025b) | 64.9 | 64.4 | 58.3 | 48.9 | 38.5 | 26.8 | 48.1 |
| | TreeVGR[‡] (Wang et al., 2025a) | 67.0 | 62.4 | 54.4 | 47.5 | **49.1** | 24.1 | 49.2 |
| | MiniO3 (Lai et al., 2025) | 74.9 | 62.8 | 57.3 | 34.4 | 36.0 | 31.7 | 48.2 |
| | *Efficiency-oriented Post Training Methods* | | | | | | | |
| | MGPO[†] (Huang et al., 2025b) | 67.5 | 62.8 | 57.3 | 44.4 | 42.0 | 29.7 | 48.7 |
| | VisionThink[‡] (Yang et al., 2025) | 61.8 | 66.9 | 60.1 | 46.6 | 39.5 | 17.9 | 45.9 |
| | **ERGO** | **81.7** | **67.1** | **66.1** | **49.6** | 43.2 | **35.0** | **55.2** |

Table 2: **Performance comparison under efficiency-considered scenarios with pixel constraints.** ERGO outperforms the original model and post-training methods across all benchmarks. † denotes reproduction with their code using our data, while ‡ denotes inference with their original pipeline.

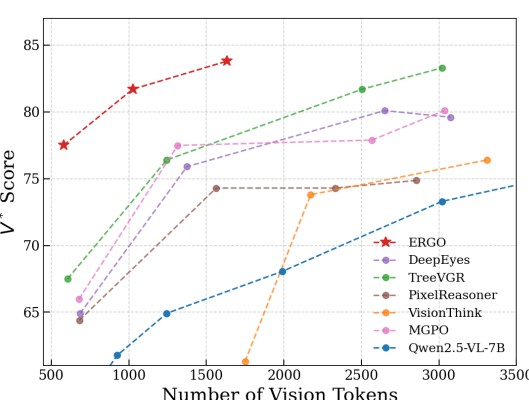

Figure 4: **Performance-efficiency trade-off on the V* benchmark.** The total number of vision tokens is the sum of the tokens from the downsampled original image and those from the high-resolution cropped image.

| Pixel Const. | Model | # of vision tokens | V* |
|---|---|---|---|
| 16384×28×28 | Qwen2.5-VL-7B | 4,471 | 77.0 |
| 1280×28×28 | PixelReasoner | 1,563 | 74.3 |
| | DeepEyes | 1,374 | 75.9 |
| | TreeVGR | 1,244 | 76.4 |
| | MGPO | 1,315 | 77.5 |
| | VisionThink | 1,749 | 73.8 |
| | MiniO3 | 1,981 | 81.2 |
| | **ERGO** | **1,632** | **83.8** |
| 640×28×28 | **ERGO** | **1,025** | 81.7 |

Table 3: **Comparison of vision token counts in coarse-to-fine reasoning.**

| Pixel Const. | Model | Max. tool cnt | V* | Latency (s) |
|---|---|---|---|---|
| 16384×28×28 | Qwen2.5-VL-7B | – | 77.0 | 4.89 |
| 640×28×28 | DeepEyes | 4 | 64.9 | 3.42 |
| | | 2 | 63.9 | 3.07 |
| | | 1 | 64.4 | 2.18 |
| 640×28×28 | MiniO3 | 4 | 74.9 | 5.35 |
| | | 2 | 61.8 | 3.87 |
| | | 1 | 41.4 | 2.03 |
| 640×28×28 | **ERGO** | 1 | **81.7** | **1.61** |

Table 4: **Latency comparison with models that leverage multiple tool calls on V* using the vLLM engine.** Latency represents the average duration to produce a final answer for each image–query pair.

image understanding. To assess potential trade-offs introduced by training, we also consider conventional multimodal benchmarks: CV-Bench (Tong et al., 2024a) and MMVP (Tong et al., 2024b) as vision-centric benchmarks; Hallusion-Bench (Guan et al., 2024), POPE (Li et al., 2023), and MMBench (Liu et al., 2024b) as general-purpose VQA tasks; and AI2D (Kembhavi et al., 2016) and ChartQA (Masry et al., 2022) for chart understanding.

## 5 RESULTS AND ANALYSIS

### 5.1 MAIN RESULTS

**High-resolution reasoning with efficient visual processing.** To measure performance under efficiency-considered scenarios, we considered two different pixel constraints—640×28×28 and

1280×28×28—corresponding to vision token limits of 640 and 1280, respectively. Since our goal is efficiency, ERGO was trained to perform reasoning with a single tool call, similar to other efficiency-oriented methods. As tool calls naturally increase latency, some methods that do not prioritize efficiency may make multiple tool calls. For a fair comparison, we constrained the maximum number of tool calls to four during all evaluations. Tab. 2 shows that, because the average resolution of input images exceeds these limits, many baseline models cannot accurately reason over images, leading to performance degradation. In contrast, ERGO consistently outperforms all baselines across benchmarks. In particular, compared to Qwen2.5-VL-7B, only ERGO achieves a higher score for every benchmark we evaluated, even under the strictest 640×28×28 pixel constraint. Qualitative results are presented in Fig. 1 and Sect. D.

Fig. 4 shows that ERGO lies in the Pareto-optimal region, achieving *higher scores with fewer vision tokens*. We evaluate ERGO with multiple pixel constraints {320, 640, 1280}×28×28, corresponding to {579, 1026, 1632} total vision tokens per sample, whereas competing baselines are evaluated at {640, 1280, 2560, 3072}×28×28. Tab. 3 also shows that with the coarse-to-fine pipeline under the 1280×28×28 constraint, ERGO achieves the highest score within the same constraint group. Remarkably, at 640×28×28, ERGO outperforms all baselines while using fewer vision tokens than others evaluated at 1280×28×28. These results demonstrate that our model achieves highly efficient utilization of the vision token, as the pixel constraints can be flexibly regulated at test-time.

**Practical latency improvements.** To demonstrate that our method not only reduces vision token usage, but also provides practical benefits in real-world deployment, we conducted a latency comparison with the original Qwen2.5-VL-7B model. The evaluation was performed using the production grade vLLM engine (Kwon et al., 2023) on a single H100 GPU with a batch size of 16, measuring the time to produce a final answer for an image-query pair. Tab. 4 shows that models leveraging multiple tool calls trade off efficiency for performance, whereas our approach achieves faster latency while simultaneously surpassing them in accuracy on the V* benchmark. See Sect. C for additional latency-performance trade-off analysis.

## 5.2 IN-DEPTH ANALYSIS

**Leveraging contextual information for VQA.** We show that the superior performance of our model arises from its ability to identify task-relevant regions even when target objects become visually indiscernible (see Fig. 1). To quantify this ability, we analyze whether the predicted region covers the GT target object, by defining Target Coverage Score.

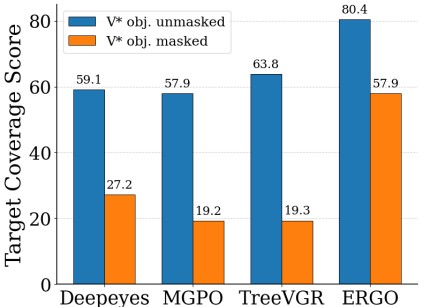

Figure 5: **Evaluation of model robustness under target-object masking.**

$$\text{Target Coverage Score} = \frac{1}{|\mathcal{B}_{gt}|} \sum_{b_g \in \mathcal{B}_{gt}} \underbrace{\max_{b_p \in \mathcal{B}_{pred}} \frac{\text{Area}(b_p \cap b_g)}{\text{Area}(b_g)}}_{\substack{\text{fraction of GT box} \\ \text{covered by best matching prediction}}}$$

where $\mathcal{B}_{pred}$ is the set of predicted bounding boxes per sample and $\mathcal{B}_{gt}$ is the set of GT bounding boxes per sample. We evaluate scores separately for cases where the object is completely masked (bounding box overlaid with black) and where it remains discernible. Since masking removes explicit visual representations of the object, models can only succeed by leveraging contextual information such as surrounding visual context or textual cues. Fig. 5 shows that ERGO achieves the most robust performance in the masked condition, consistent with its stronger ability to exploit such contextual signals.

**Bias-free region prediction with the box adjustment constant.** We employ a fixed box adjustment constant (i.e., $\gamma$ is used in $r_{box}$ of Eq. 2) to facilitate efficient training; in principle, the model could be guided to predict the largest region permitted by the constant. However, Fig. 6 shows that ERGO infers regions with flexible areas that reflect the underlying characteristics of the data: in MMVP (Tong et al., 2024b), objects often occupy the full frame, whereas in MME-RWL (Zhang et al., 2024), objects are relatively small. This indicates that the box adjustment constant does not bias ERGO toward fixed-size predictions.

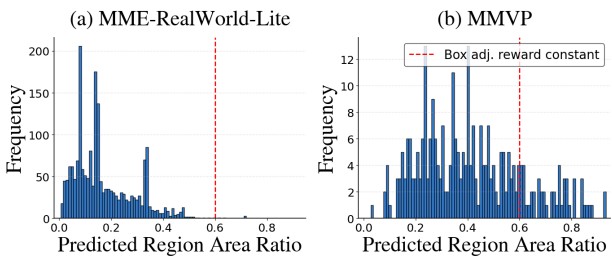

Figure 6: **Bias-free region prediction.** ERGO adapts region sizes properly for (a) the high-resolution MME-RWL and (b) the low-resolution MMVP, indicating that the box adjustment constant does not bias the region predictions.

| Benchmark | Qwen2.5-VL | **ERGO** |
|---|---|---|
| CVBench-2D | 74.1 | **76.0** |
| CVBench-3D | 73.0 | **80.3** |
| MMVP | 77.0 | **77.7** |
| Hallusion-Bench | 47.1 | **52.3** |
| POPE | 86.4 | **87.4** |
| MMBench | 82.1 | **82.9** |
| AI2D | 81.3 | **84.7** |
| ChartQA | **86.1** | 85.8 |

Table 5: **Results on conventional vision–language benchmarks.** ERGO maintains or improves the capabilities of the base Qwen2.5-VL-7B model.

| No. | Method | $r_{acc}$ | $r_{region}$ | $r_{box}$ | RW | Avg. |
|---|---|---|---|---|---|---|
| | Qwen2.5-VL-7B | | | | | 52.4 |
| Ⓐ | $r_{acc}$ *only* | ✓ | | | | 53.5 |
| Ⓑ | $r_{region}$ *only* | | ✓ | | | 51.4 |
| Ⓒ | +box adj. reward | ✓ | ✓ | | | 54.9 |
| Ⓓ | +reward weighting (RW) | ✓ | ✓ | ✓ | | 55.3 |
| Ⓔ | ***ERGO*** | ✓ | ✓ | ✓ | ✓ | **58.4** |

(a) Reward design

| No. | $(\alpha, \beta)$ | Avg. |
|---|---|---|
| (i) | (1.0, 1.0) | 56.2 |
| (ii) | (1.0, 2.0) | 54.3 |
| (iii) | (0.5, 0.25) | 56.8 |
| (iv) | (1.0, 0.5) | **58.4** |

(b) TCE reward weight

| Parameter size | Average |
|---|---|
| 3B | 55.6 |
| 7B | 56.4 |
| 72B | **58.4** |

(c) Parameter size

| Reward model | Average |
|---|---|
| GLM4.5V-108B | 58.2 |
| InternVL3-78B | 57.2 |
| Qwen2.5-VL-72B | **58.4** |

(d) Reward model

| $\gamma$ | Function | Average |
|---|---|---|
| 0.4 | step | 57.8 |
| 0.8 | step | 56.2 |
| 0.6 | step | **58.4** |
| 0.6 | linear anneal. | 51.0 |

(e) Box adjustment reward

Table 6: **Ablation analysis.** Average performance is measured over six benchmarks in Tab. 2.

**Results on conventional multimodal benchmarks.** We evaluated ERGO on a broad set of multimodal benchmarks, including general VQA, vision-centric VQA, and document understanding. As shown in Tab. 5, ERGO not only maintains the abilities of the base model but also achieves improvements on several benchmarks. We attribute these gains to the improved ability of the model to reason in semantically relevant regions.

## 5.3 ABLATION STUDIES

**The TCE reward is more effective than the accuracy reward.** In Tab. 6(a), Ⓓ relies solely on the TCE reward, without generating task answers during training, whereas Ⓐ relies solely on the accuracy reward, without evaluating the quality of the cropped region. Remarkably, although the final performance is measured by answer accuracy, Ⓓ—which was never explicitly trained to answer the task—still outperforms Ⓐ. The results highlight the effectiveness of the TCE reward design, because improving the quality of the selected region with task-relevant evidence is critical to performance in the coarse-to-fine pipeline.

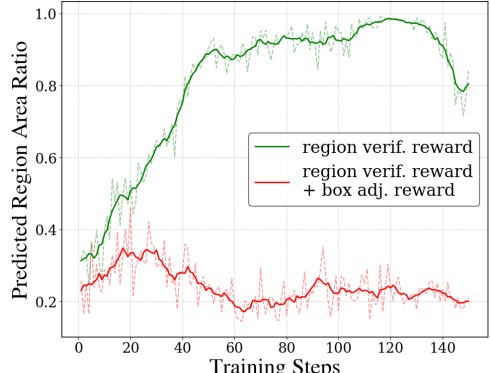

Figure 7: **Impact of box adjustment reward on predicted regions during training.**

**The box adjustment reward is critical for effective training.** In Fig. 7, removing the box regularization reward drives the model toward the trivial policy of cropping overly large regions. Evidenced

by the superior performance of Ⓒ compared to Ⓑ in Tab. 6(a), the removal of the box adjustment reward not only causes inefficient inference but also impairs effective model training.

**Regularizing the prioritization of the box adjustment reward is beneficial.** For Ⓒ in Tab. 6(a), we set $\alpha = 1$ and $\beta = 1$ in Eq. 3. For Ⓓ, we reduced the weight of the box adjustment reward to $\beta = 0.5$ to prevent the policy from overly prioritizing this term over more critical rewards. This coordination of the weights results in higher average scores, which confirms our intended effect. In Tab. 6(b), we further investigate multiple configurations of $\alpha$ and $\beta$. Configuration (ii), where the box-adjustment weight $\beta$ is larger than $\alpha$, shows the lowest performance. In contrast, configuration (iii), which scales both $\alpha$ and $\beta$ by a common multiplicative factor while preserving their relative ratio, outperforms configuration (i) even when the region verification signal is weakened. This further underscores the strength of our TCE reward formulation and validates the proposed weighting strategy.

**The accuracy reward is complementary to the TCE reward.** In Tab. 6(a), while Ⓐ underperforms compared to Ⓒ, combining the accuracy reward with the TCE reward can yield benefits during training. This complementary effect is particularly valuable when the model successfully selects the task-relevant region but struggles to answer from the cropped sub-image. By leveraging both rewards, ERGO with Ⓔ can address both the quality of the cropped image and the training-test mismatch, thereby enhancing overall performance.

**Our reward design plays a critical role regardless of size of the reward model and the model family.** We evaluate the impact of reward model size by varying Qwen2.5-VL-Instruct (Bai et al., 2025) from 72B to 7B and 3B. As shown in Tab. 6(c), the performance does not collapse but remains robust, exhibiting only limited sensitivity to the model scale. Furthermore, Tab. 6(d) shows that performance remains stable across different reward model families. We attribute this robustness to the nature of our proxy task for assessing cropped image quality. Evaluating a reward model's response given an image crop and a query is a fundamentally straightforward task, enabling even lower-capacity reward models to perform it reliably.

**Data-driven selection of a $\gamma$ is effective.** Tab. 6(e) supports our strategy for selecting the box adjustment constant $\gamma$ in Eq. 2. By setting $\gamma$ based on the data statistics, we effectively guide the model's training behavior. A large $\gamma$ would fail to penalize the trivial solution of cropping excessively large regions. Conversely, a small $\gamma$ would restrict the model's ability to explore and identify the most relevant visual areas. Our data-driven approach strikes a balance, encouraging focused exploration while maintaining training stability.

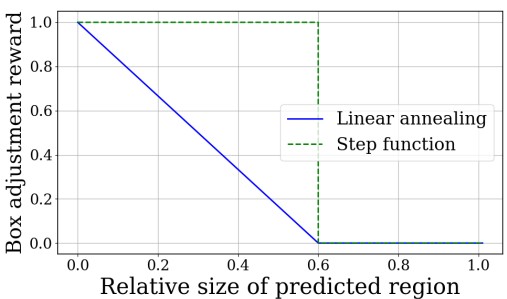

Figure 8: **Reward function for box adjustment reward.**

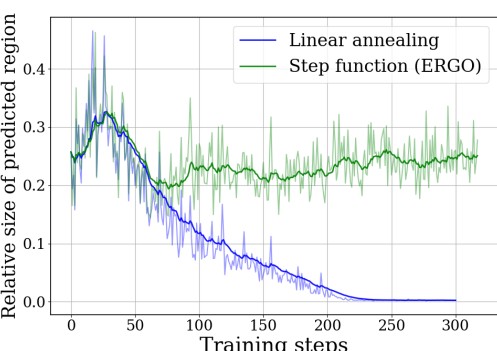

Figure 9: **Impact of reward functionon predicted regions during training.**

**Objective-aligned box adjustment reward degrades training performance.** We experimented with a linear annealing function to gradually emphasize smaller region selections, as shown in Fig. 8, which aligns with our efficiency goal of reducing vision tokens. However, our empirical results indicate that annealing degrades performance: the policy becomes overly incentivized to shrink predicted regions, as illustrated in Fig. 9, because reducing region size is substantially easier than improving the region-verification reward. Consequently, Tab. 6(e) shows that the model trained with linear annealing fails to reason over semantically meaningful areas.

## 6 RELATED WORK

**LVLMs reasoning on vision spaces.** The remarkable reasoning capabilities of RL post-trained Large Language Models (LLMs) have significantly advanced problem-solving. Approaches such as GRPO (Shao et al., 2024b) demonstrate that grouped reward signals can effectively induce complex reasoning. Building on these advancements in text-only LLMs, substantial efforts have been made to extend similar reasoning schemes to LVLMs. Early efforts (Shen et al., 2025; Huang et al., 2025a) integrated vision inputs for reasoning, using GRPO-like techniques to improve LVLM reasoning with text-only exploration. More recently, the concept of "thinking with images" (Su et al., 2025b), exemplified by models such as OpenAI-o3 (OpenAI, 2025), has gained traction, emphasizing visual-space reasoning. While reasoning LVLMs (Zheng et al., 2025b; Wang et al., 2025a) have been widely studied to boost performance, their use for efficient inference remains under-explored. Our work addresses this by showing that ERGO with grounded region supervision can achieve both higher efficiency and greater task-solving ability.

**Efficient LVLMs with vision token pruning.** The efficiency bottleneck lies in the rapid growth of vision token count as input image resolution increases. Vision token pruning (Chen et al., 2024; Wen et al., 2025b; Lee et al., 2025) mitigates this by selectively removing tokens to reduce computation. However, these methods often rely on layer-specific inference schemes, making them unsuitable for production-grade engines (Kwon et al., 2023; Zheng et al., 2024) that lack support for dynamic sequence lengths across layers. As noted by Wen et al. (2025a), such pruning often yields theoretical FLOPs reductions, which rarely translate into real inference-time speedups. Their focus is largely on compensating accuracy loss rather than achieving performance gains. In contrast, ERGO provides both performance gains and practical latency improvements within production-grade LLM engines.

**Efficient LVLMs with RL.** RL has been explored as a method to improve the efficiency of LVLMs. While moderating image resolution is a straightforward approach, it comes with the trade-off of reducing visual information. To address this, some RL-trained methods empower the model to manage resolution itself. For instance, VisionThink (Yang et al., 2025) trains models to request higher resolution when an image is too ambiguous to answer a question. However, this approach remains redundant, as it reprocesses the entire image at a higher resolution rather than focusing on task-relevant regions. In contrast, MGPO Huang et al. (2025b) trains models with downsampled images and high-resolution cropped regions, rewarding final answer accuracy. However, by neglecting the quality of the selected regions, MGPO fails to surpass methods without an efficiency objective. By assessing predicted regions with efficiency-oriented objective, ERGO achieves the best efficiency in high-resolution visual understanding.

## 7 CONCLUSION

Our study reveals a critical limitation of existing *perception-driven reasoning* models: their performance substantially degrades under low-resolution inputs in coarse-to-fine reasoning scenarios. These models rely heavily on clearly discernible visual anchors to localize objects; when such cues are lost due to downsampling, their ability to identify task-relevant regions deteriorates, causing errors in reasoning and question answering. This underscores the need for approaches that capture coarse cues while selectively attending to semantically salient regions. Our **ERGO** conducts *reasoning-driven perception*, maintaining both efficiency and accuracy even when high-fidelity object information is lost, thereby overcoming the efficiency shortcomings of prior methods.

ACKNOWLEDGMENTS

This research was supported by Artificial intelligence industrial convergence cluster development project funded by the Ministry of Science and ICT (MSIT, Korea) & Gwangju Metropolitan City.

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

# Appendix — ERGO: Efficient High-Resolution Visual Understanding For Vision-Language Models

## A  TRAINING ALGORITHM

---

**Algorithm 1:** Policy updates with our reward design

---

**Input:** Policy model $\pi_\theta$, reward model $\mathcal{R}$, train set $\{(I_{\text{orig},i}, q_i, o_{\text{GT},i})\}_{i=1}^N$, group size $G$, reward weights $\alpha, \beta$, box adjustment constant $\gamma$, stability constant $\epsilon$

**while** *training* **do**

    **foreach** *sample* $(I_{orig}, q, o_{\text{GT}})$ *in the train set* **do**

        Initialize empty lists: GroupRewards $\leftarrow$ [], GroupRollouts $\leftarrow$ []

        **foreach** *rollout* $g = 1, \ldots, G$ **do**

            $o_{\text{region}} \sim \pi_\theta(\cdot | I_{\text{orig}}, q)$

            **if** *$o_{region}$ is not valid for crop* **then**

                `// e.g.  impossible to parse out the bounding-box,`

                `// missing value at the coordinates, etc.`

                $r_{\text{TCE}}, r_{\text{acc}}, r_{\text{format}} \leftarrow 0$

            **else**

                $I_{\text{region}} \leftarrow \text{crop}(I_{\text{orig}}, o_{\text{region}})$

                $o_{\text{acc}} \sim \pi_\theta(\cdot | [I_{\text{region}}, q], [I_{\text{orig}}, o_{\text{region}}])$

                $o_{\text{RM}} \sim \mathcal{R}(\cdot | I_{\text{region}}, q)$

                $r_{\text{region}} \leftarrow \mathbb{1}[\text{match}(o_{\text{RM}}, o_{\text{GT}})]$

                $r_{\text{box}} = \mathbb{1}[\frac{\text{Area}(I_{\text{region}})}{\text{Area}(I_{\text{orig}})} \leq \gamma]$

                $r_{\text{acc}} \leftarrow \mathbb{1}[\text{match}(o_{\text{acc}}, o_{\text{GT}})]$

                $r_{\text{format}} \leftarrow \mathbb{1}[\, o_{\text{region}}, o_{\text{acc}} \text{ follow expected format}\,]$

                `// Task-driven Contextual Exploration (TCE) Reward`

                $r_{\text{TCE}} = \alpha \cdot r_{\text{region}} + \beta \cdot r_{\text{box}}$

            $R \leftarrow r_{\text{TCE}} + r_{\text{acc}} + r_{\text{format}}$

            Append $(o_{\text{region}}, o_{\text{acc}})$ to GroupRollouts

            Append $R$ to GroupRewards

        $\bar{R} \leftarrow \frac{1}{G} \sum_{g=1}^G \text{GroupRewards}[g]$

        $\sigma_R \leftarrow \sqrt{\frac{1}{G} \sum_{g=1}^G (\text{GroupRewards}[g] - \bar{R})^2}$

        Advantages $\leftarrow \left\{ \frac{R_g - \bar{R}}{\epsilon + \sigma_R} \right\}_{g=1}^G$ for each $R_g \in$ GroupRewards

    `// Policy update following GRPO (Shao et al., 2024b)`

    $\pi_\theta \leftarrow$ update $\pi_\theta$ using GroupRollouts and Advantages

**Output:** Learned policy model $\pi_\theta$

---

## B  TRAINING DETAILS

| Parameter | Value |
|---|---|
| Base model | Qwen2.5-VL-7B-Instruct |
| Data | V* training set, ArxivQA |
| Hardware | NVIDIA H100 |
| Optimizer | AdamW |
| Total training steps | 250 |
| Global batch size | 128 |
| Rollouts per sample | 8 |
| Learning rate | $1 \times 10^{-6}$ |
| RL algorithm | GRPO |
| Reward model | Qwen2.5-VL-72B-Instruct |
| GPU hours | ~150 |

Table 7: Training configuration.

Table 7 summarizes the training setup.

**Models.** We adopted Qwen2.5-VL-7B-Instruct (Bai et al., 2025) as the base model for RL training, owing to its strong vision–language reasoning ability and object-level referring detection, which enable effective grounding without cold-start initialization. Moreover, Qwen2.5-VL has been widely used in prior RL-based studies, ensuring fair comparison with related work. For the reward model, we used Qwen2.5-VL-72B-Instruct, one of the most powerful open-sourced LVLMs, to provide a reliable and precise reward signal.

**Data.** We followed the setup of DeepEyes (Zheng et al., 2025b), reusing their curated training data for RL post-training. This choice isolates the contribution of our method from dataset curation effects, allowing us to demonstrate improvements independently of data filtering, though such filtering remains a valid and complementary approach.

**Other training details.** Training was performed on a cluster node with 4 H100 GPUs. The global batch size was 128. For accuracy rewards, half of each mini-batch was allocated to longer rollouts to avoid VRAM bottlenecks. Sixteen rollouts were sampled per training example. The learning rate was fixed at $1 \times 10^{-6}$ throughout training. We employed standard GRPO, as alternative variants such as DR.GRPO Liu et al. (2025) and GSPO (Zheng et al., 2025a) did not yield significant improvements in preliminary trials.

# C   ANALYZING LATENCY−PERFORMANCE TRADE-OFFS BEYOND FIXED PIXEL CONSTRAINTS

We conducted additional experiments to broaden the scope of evaluation. Beyond the fixed pixel-constraint setting, we evaluated two supplementary comparisons: one under performance-constrained settings, where each model is evaluated using its native configuration, and another under latency-constrained settings, where all models operate within comparable inference-time limits. These additional tables provide a more complete view of ERGO's performance across different practical constraints. All experiments are conducted using the vLLM engine (Kwon et al., 2023) on a single H100 GPU with a batch size of 16.

| Model | V* | Latency (s) | Remarks |
|---|---|---|---|
| TreeVGR | 85.3 | 5.3 | Original setting |
| DeepEyes | 84.3 | 9.4 | Original setting |
| Mini-o3 | 86.8 | 9.0 | Original setting |
| ERGO | 85.9 | 5.1 | Pixel constraints: $2256 \times 28 \times 28$ |
| ERGO | 83.8 | 2.6 | Pixel constraints: $1280 \times 28 \times 28$ |

Table 8: Comparison under original baseline settings without restricting baselines' configurations.

Table 8 shows the results under the native configurations of the models. When computational limits are relaxed (e.g., through larger pixel budgets), ERGO matches or exceeds the performance of stronger but slower baselines, while maintaining lower latency.

| Model | V* | Latency (s) | Remarks |
|---|---|---|---|
| TreeVGR | 74.4 | **2.6** | Pixel constraints: $2048 \times 28 \times 28$ |
|  | 81.7 | 3.1 | Pixel constraints: $2560 \times 28 \times 28$ |
| DeepEyes | 64.9 | 3.2 | Pixel constraints: $640 \times 28 \times 28$ |
|  | 75.9 | 3.8 | Pixel constraints: $1280 \times 28 \times 28$ |
| Mini-o3 | 74.4 | **2.6** | Single tool call; Pixel constraints: $1280 \times 28 \times 28$ |
|  | 74.4 | 5.0 | Single tool call |
|  | **83.8** | 7.0 | Up to 6 tool calls |
| ERGO | **83.8** | **2.6** | Pixel constraints: $1280 \times 28 \times 28$ |

Table 9: Performance under matched latency budgets by regularating pixel constraints or tool call counts.

To evaluate models in latency-sensitive environments, we further cap inference time per sample and tune pixel budgets or permissible tool-call counts accordingly. As shown in Table 9, ERGO achieves the highest accuracy among all models when operating under comparable latency budgets, demonstrating its efficiency−performance advantage in constrained settings.

# D    QUALITATIVE RESULTS

The qualitative results indicate that ERGO effectively leverages coarse visual cues when needed.

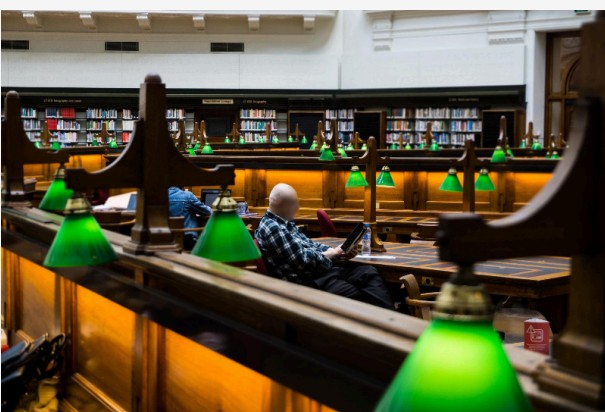

*Resized from original image (Pixel Constraint : 1280×28×28)*

**Question**

What is the color of the bottle cap?
First, consider how to zoom in on the image to include only the region containing all the information necessary to answer the question.

**Reasoning**

To determine the color of the bottle cap, I need to focus on the region where the bottle is located. The bottle is visible on the table in front of the person wearing a plaid shirt.

Zoom in on the table in front of the person wearing a plaid shirt to focus on the bottle

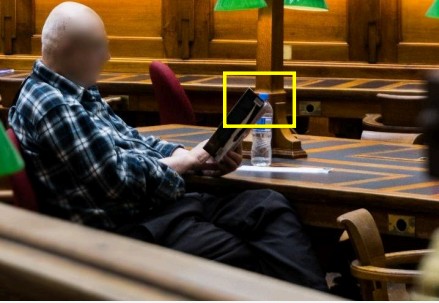

*Cropped at original image (Pixel Constraint : 1280×28×28)*

**Question**

What is the color of the bottle cap?

**Reasoning**

The bottle cap is located on the water bottle next to the person reading. Upon zooming in, the cap appears to be blue.

**Answer**

Blue.

Figure 10: **ERGO utilizes coarse cues ("the region where the bottle is located") to provide the answer.** The yellow box highlights the region linked to the answer for clear visualization.

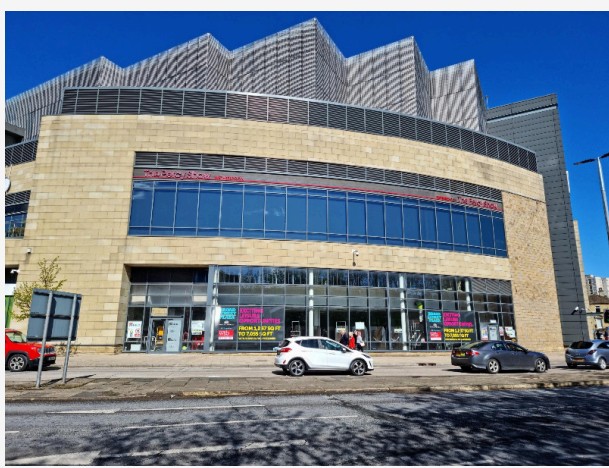

*Resized from original image (Pixel Constraint : 1280×28×28)*

**Question**

What is the color of the pink-haired woman's handbag?
First, consider how to zoom in on the image to include only the region containing all the information necessary to answer the question.

**Reasoning**

To determine the color of the pink-haired woman's handbag, I need to focus on the area near the entrance of the building where people are visible. The region around the entrance contains the necessary information.

Zoom in on the entrance area to focus on the people near the building.

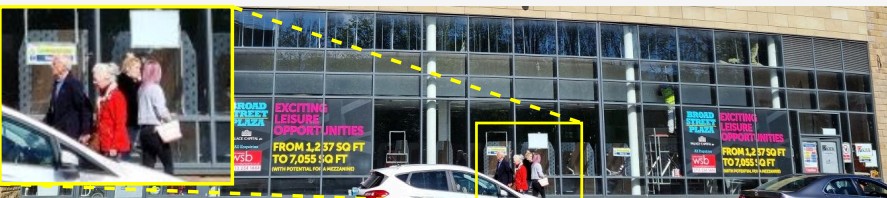

*Cropped at original image (Pixel Constraint : 1280×28×28)*

**Question**

What is the color of the pink-haired woman's handbag?

**Reasoning**

The enlarged sub-image focuses on the entrance area where a person with pink hair is visible. The handbag she is carrying appears to be white.

**Answer**

White.

Figure 11: **ERGO leverages coarse cue ("where people are visible") to answer the question.** The yellow box indicates the region associated with the answer to aid visualization.

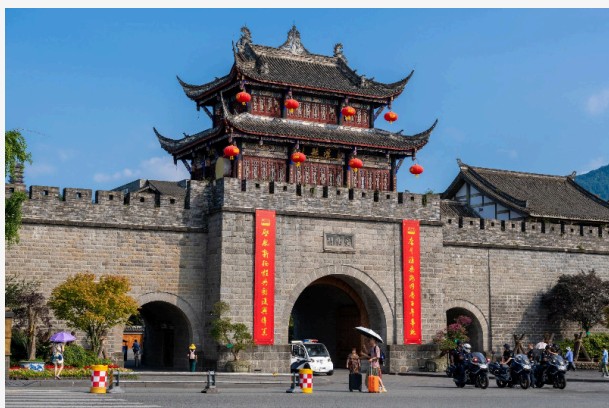

*Resized from original image (Pixel Constraint : 1280×28×28)*

**Question**

Is the orange luggage on the left or right side of purple umbrella?
First, consider how to zoom in on the image to include only the region
containing all the information necessary to answer the question.

**Reasoning**

To determine whether the orange luggage is on the left or right side of the
purple umbrella, I need to focus on the area around the umbrella and the
luggage. The umbrella is located near the center-left of the image, and the
orange luggage is visible to the right of the umbrella.

Zoom in on the area around the purple umbrella and the orange luggage.

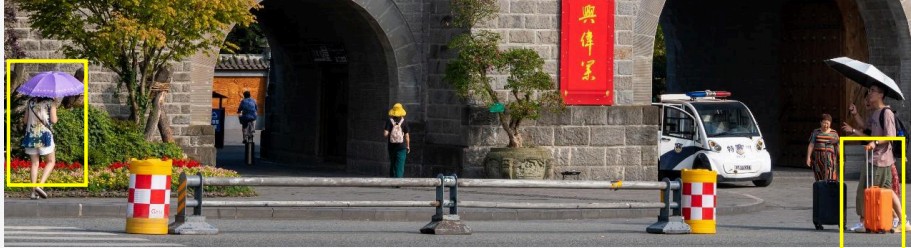

*Cropped at original image (Pixel Constraint : 1280×28×28)*

**Question**

Is the orange luggage on the left or right side of the purple umbrella?

**Reasoning**

The enlarged sub-image focuses on the area around the purple umbrella and
the orange luggage, making it clear that the orange luggage is positioned to the
right of the umbrella.

**Answer**

Right.

Figure 12: **ERGO can also exploit clear visual cues (the purple umbrella and the orange luggage) when the object is still discernible.** The yellow box highlights the region associated with the answer for clear visualization.

# E    LLM USAGE DISCLOSURE

This paper utilized a large language model (LLM) solely for the purpose of checking grammar, spelling, and typographical errors.

