# OpenReview forum: "ERGO: Efficient High-Resolution Visual Understanding for Vision-Language Models"
_ICLR.cc/2026/Conference — ICLR 2026 Poster_

### Official Review · Reviewer_9PNo · 2025-10-26

**Soundness:** 2
**Presentation:** 2
**Contribution:** 2
**Rating:** 4
**Confidence:** 3

**Summary:**

Directly processing high-resolution images increases computational overhead, as many processed tokens are not task-relevant. To address this, the paper proposes a two-stage, coarse-to-fine approach. In the first stage, the model processes a low-resolution image to identify task-relevant regions; in the second stage, cropped high-resolution images of these regions are used for further processing. To enable this mechanism, the authors combine standard accuracy and format rewards with a box adjustment reward and a region-verification reward. As a result, the model achieves better performance than the base model while processing significantly fewer image tokens.

**Strengths:**

1. The paper is well motivated, and the two-stage coarse-to-fine approach makes sense and appears effective.

2. The model performs better than the base model while processing fewer image tokens.

3. The motivation for the additional rewards, such as the region-verification and box adjustment rewards, is clear.

**Weaknesses:**

The system appears complex due to the use of multiple rewards and reward models.

There is no ablation study on the weights of the different rewards. For example, in Line 216, the paper directly specifies the weights without any explanation or ablation analysis.

Some experimental details seem unclear. For instance, the paper lists DeepEyes [1] as a baseline for comparison, but the reported evaluation metrics differ significantly from those in the original DeepEyes paper.

Reference:
[1] DeepEyes: Incentivizing “Thinking with Images” via Reinforcement Learning

**Questions:**

What causes the significant drop in DeepEyes’ performance as reported in the table 2?

---

> ### Author Response · Authors · 2025-11-18
> **Author Rebuttal to 9PNo’s Comments  (1/N)**
>
> > W1. Clarification of system complexity from reward designs
> - We acknowledge the concern regarding system complexity arising from the use of multiple rewards and reward models. However, each reward component is well-motivated and empirically justified. As shown in the ablation results in **Table 6 (and the table below)**, both the region-verification and box-adjustment rewards are essential for strong performance. Our full reward design achieves an average accuracy of **58.4%** across six benchmarks, compared to **53.5%** with only a simple accuracy reward. Although simplicity is desirable, overly simplified reward formulations lead to notable performance drops, making our proposed design both necessary and effective.
>   | No. | Method              | r_acc | r_region | r_box | RW  | Avg. |
>   |-----|---------------------|:-----:|:--------:|:-----:|:---:|-----:|
>   | Ⓐ   | r_acc only          |   ✔   |          |       |     | 53.5 |
>   | Ⓔ   | **ERGO**            |   ✔   |    ✔     |   ✔   |  ✔  | **58.4** |
> - Furthermore, our reward design substantially reduces data requirements. Our RL-only approach uses only question–answer (Q&A) pairs, whereas supervised finetuning (SFT) baselines rely on more labor-intensive annotations such as bounding boxes and, in many cases, reasoning-path text. This reduced dependence on complex supervision makes our approach far easier to scale. Moreover, despite using far less annotated data, our RL-only method still outperforms SFT and SFT→RL baselines by a substantial margin, demonstrating the effectiveness of our reward design, as shown in the results below and the updated manuscript (Table 2).
>
> | Model   | Training Method | V* Bench | HR Bench 4K | HR Bench 8K | MME-RW Lite | TreeBench | VisualProbe | Average |
> |------------------|------------------------------|---------:|------------:|------------:|-------------:|----------:|------------:|--------:|
> | Qwen2.5-VL-7B-Inst. | zero shot |  56.5  |65.6 |  56.5 |  42.6  |  40.0  |  15.2  |   **44.8**  |
> | Vis-CoT | only SFT |  56.5  | 46.6 |  40.8 |  35.6  |   30.6  |  23.5  |   **38.0**  |
> | Chain-of-Spot | only SFT |  48.4  | 41.1 |  36.5 | 31.6  | 32.1  | 18.1  |   **33.8**  |
> | PixelReasoner | SFT -> RL |    74.5  |        66.9 |        61.3 |        49.8  |     40.4  |       31.8  |   **52.1**  |
> | TreeVGR    | SFT -> RL |  76.4  |        66.4 |        60.4 |        47.5  |   48.2 |       26.6  |   **53.1**  |
> |Mini-o3     | SFT -> RL |  81.2 |       70.0 |        65.9 |        36.7  |     36.3  |       39.0  |   **52.2**  |
> |MGPO      | only RL |  77.5  |        69.8 |        61.1 |        44.4  |     39.3  |       33.4  |   **52.6**  |
> | DeepEyes   | only RL |  78.5  |        66.0 |        60.0 |        48.9  |     40.2  |       39.0  |   **52.5**  |
> | **ERGO**  | only RL  | 83.8 |     73.0 |     69.9 |     52.6 |   41.7 |   42.7 | **58.4** |
> ##### Pixel Const.: 1280×28×28
>
> ---
>
> > W2. Ablation study on the weighting of different rewards
> - Our final reward is the sum of the Task-driven Contextual Exploration (TCE) reward and conventional rewards (accuracy and format rewards), using a **1:1:1 weighting**. We keep these weights fixed for simplicity, and this setting has been shown to be empirically effective in our experiments.
>
> - Within the TCE reward, we use α = 1 for the region-verification reward and β = 0.5 for the box-adjustment reward. **Table 6(a)** in our paper already includes ablations on these weightings:
>
>   - Ⓑ: region-verification only (α = 1, β = 0),
>
>   - Ⓒ: region-verification + box-adjustment with equal weighting (α = 1, β = 1),
>
>   - Ⓓ: region-verification + box-adjustment with the proposed weighting (α = 1, β = 0.5).
>
> - As shown in **the table below and the revised manuscript (Table 6(b))**, we further investigate multiple configurations of $\alpha$ and $\beta$. Configuration (ii), where the box-adjustment weight $\beta$ is larger than $\alpha$, shows the lowest performance. In contrast, configuration (iii), which scales both $\alpha$ and $\beta$ by a common multiplicative factor **while preserving their relative ratio, outperforms configuration (i)** even when the region verification signal is weakened. This further **underscores the strength of our TCE reward** formulation and validates the proposed weighting strategy.
>
>   | No.  | (α, β)       | Avg.  |
>   |:----:|:------------:|:-----:|
>   | (i)  | (1.0, 1.0)   | 56.2  |
>   | (ii) | (1.0, 2.0)   | 54.3  |
>   | (iii)| (0.5, 0.25)  | 56.8  |
>   | (iv) | **(1.0, 0.5)** | **58.4** |

---

> > ### Author Response · Authors · 2025-11-18
> > **Author Rebuttal to 9PNo’s Comments (2/N)**
> >
> > > W3 & Q1. Clarification of the performance of DeepEyes
> > - The performance of DeepEyes appears lower in our results because we **evaluate all models under a strict resolution-constrained**, efficiency-oriented setting, which differs from the conditions used in the original DeepEyes paper. Under these pixel constraints, task-relevant objects often become indistinguishable in the low-resolution input, and we found that perception-driven reasoning models like DeepEyes are not robust in such environments. Their performance drops sharply when the target region is difficult to perceive at low resolution. Therefore, the gap in DeepEyes’ reported performance arises from differences in evaluation conditions rather than inconsistencies in implementation.
> > - We also emphasize that our work, based on reasoning-driven perception, specifically targets efficient high-resolution reasoning under strict pixel budgets, and the evaluation protocol in **Table 2** reflects this efficiency-oriented setup.

---

> ### Comment · Reviewer_9PNo · 2025-11-27
>
> Thanks to the authors for the clarification.
>
> I question the fairness of a strictly resolution-constrained evaluation for these benchmarks and models, which are intended for high-resolution images. Although I appreciate the motivation for efficient high-resolution inference, lowering the baseline performance by changing the experimental setting seems inappropriate, as the models were not designed for this lower-resolution task.
>
> A more equitable approach is to compare all models using their original settings and then analyze the computational expense. The claimed benefits of ERGO would be justified if it achieves comparable performance with a lower computational budget.
>
> For these reasons, I intend to maintain my original score, but would not mind if the paper is accepted.

---

> ### Author Response · Authors · 2025-11-27
> **Author Rebuttal to 9PNo’s Comment**
>
> Thank you for your feedback and for the thoughtful comments. We deeply appreciate your comment and hope that we could further get the response from our additional results.
>
> > Fairness considerations.
>
> We agree that a strictly resolution-constrained evaluation can be challenging for models originally designed for high-resolution inputs. Our primary contribution includes both **coarse-to-fine reasoning and a reward design that is naturally suited to this two-stage process**. Therefore, it was important to examine whether existing open-source baselines can readily adapt to this framework. Our experiments suggest that they cannot. These baselines **do not easily accommodate coarse-to-fine reasoning**. Evaluating them under this setting is crucial, because if models trained without our RL-based reward design performed well in this context, it would significantly reduce the significance of our contribution.
>
> At the same time, we deeply understand your concern about fairness when models are not evaluated under their original configurations. To address this, we conducted additional experiments and two new tables: **one under performance-constrained settings and another under latency-constrained settings** in addition to the original fixed pixel-constraints evaluation.
>
> -----
>
> > An Experiment without compromising the original settings of baselines.
>
> Table shows that ERGO matches the performance of competing models, when given larger computational constraints, while still maintaining the fastest latency.
>
> | Models    | V*   | V* (All A) | Latency (s) | Remarks                                         |
> |-----------|------|------------|-------------|------------------------------------------------|
> | ERGO      | 83.8 | 86.4       | 2.6        | Original setting |
> | ERGO      | 85.9 | 88.0       | **5.1**         | Regulate the pixel constraints to **2256×28×28** |
> | TreeVGR   | 85.3 | 88.5       | 5.3         | **Original setting**                        |
> | DeepEyes  | 84.3 | 88.0       | 9.4         | **Original setting**                        |
> | Mini-o3   | 86.8 | 88.4       | 9.0         | **Original setting**                     |
>
> -----
>
> > An experiment with latency constraints for further comparison.
>
> Table caps the per-sample latency. ERGO achieves the highest performance under comparable latency budgets.
>
> | Models    | V*   | V* (All A) | Latency (s) | Remarks                                                      |
> |-----------|------|------------|-------------|-------------------------------------------------------------|
> | ERGO      | **83.8** | **86.4**       | 2.6         | Original setting                                                     |
> | TreeVGR   | 74.4 | 79.1       | 2.6         | Regulate the pixel constraints to 2048×28×28  |
> | TreeVGR   | 81.7 | 84.3       | 3.1         | Regulate the pixel constraints to 2560×28×28 |
> | DeepEyes  | 64.9 | 71.2       | 3.2         | Regulate the pixel constraints to 640×28×28  |
> | DeepEyes  | 75.9 | 80.1       | 3.8         | Regulate the pixel constraints to 1280×28×28  |
> | Mini-o3   | 74.4 | 74.9       | 2.6         |Regulate to the single tool call & pixel onstraints to 1280×28×28     |
> | Mini-o3   | 74.4 | 74.4       | 5.0         | Regulate to the single tool call             |
> | Mini-o3   | 83.8 | 85.9       | 7.0         | Regulate to the maximum 6 tool call
>
> -----
> > Explanation for V*(All A)
>
> We also want to clarify an important detail regarding V* evaluation. Prior works have used two different versions of the benchmark:
>
> 1. The original V* question–option pairs released with the dataset.
> 2. An alternative version from the V* image annotation files, where all answers are mapped to option “A,” which generally yields higher scores.
>
> The official V* authors, as well as widely used evaluation toolkits such as lmms-eval [1] and VLMEvalKit [2], follow the first format. All results in our paper and Mini-o3 [3] use this standard. DeepEyes[4] and TreeVGR [5] based on their open-sourced repository, use the second “all-A options” version. For completeness and fairness, we additionally tested ERGO on this variant as well.
>
> -----
>
> ###### [1] Li, Bo, et al. LMMs-Eval: Accelerating the Development of Large Multimodal Models. Version 0.1.0, Zenodo, Mar. 2024, https://github.com/EvolvingLMMs-Lab/lmms-eval
> ###### [2] Duan, Haodong, et al. "Vlmevalkit: An Open-Source Toolkit for Evaluating Large Multi-Modality Models." Proceedings of the 32nd ACM International Conference on Multimedia, 2024, pp. 11198–11201.
> ###### [3] Lai, Xin, et al. "Mini-o3: Scaling up reasoning patterns and interaction turns for visual search." arXiv preprint arXiv:2509.07969 (2025).
> ###### [4] Zheng, Ziwei, et al. "DeepEyes: Incentivizing 'Thinking with Images' via Reinforcement Learning." arXiv preprint arXiv:2505.14362, (2025).
> ###### [5] Wang, Haochen, et al. "Traceable evidence enhanced visual grounded reasoning: Evaluation and methodology." arXiv preprint arXiv:2507.07999 (2025).

---

### Official Review · Reviewer_uFQT · 2025-10-28

**Soundness:** 3
**Presentation:** 3
**Contribution:** 3
**Rating:** 6
**Confidence:** 4

**Summary:**

This paper introduces ERGO (Efficient Reasoning & Guided Observation), a two-stage "coarse-to-fine" framework aimed at improving the efficiency of high-resolution image understanding in Large Vision-Language Models (LVLMs). While existing two-stage methods often fail when target objects are indiscernible in low-resolution inputs, ERGO solves this by transitioning from "perception-driven reasoning" to "reasoning-driven perception," which uses contextual information to locate relevant image regions.
To achieve this, the authors employ a reinforcement learning (RL) scheme and propose the Task-driven Contextual Exploration (TCE) reward. This reward includes a component that ensures the cropped region is self-sufficient for answering the query, and another that actively encourages smaller, more efficient crops. Experiments on high-resolution benchmarks, including V* and HR-Bench, show that ERGO achieves superior accuracy while drastically reducing the number of vision tokens used, leading to faster inference compared to the base model and competing efficient methods.

**Strengths:**

1. Solves a Real and Critical Bottleneck: The paper tackles a major pain point: the prohibitive cost of processing high-resolution images for LVLMs in real-world applications. The authors sharply identify the failure case of existing coarse-to-fine methods—where tiny objects disappear after downsampling, and offer an effective, targeted solution.

2. Strong Experimental Results and Efficiency Gains: ERGO not only achieves sota performance on several high-resolution benchmarks like V* but also dramatically cuts down the computational cost (fewer Tokens, less latency).

**Weaknesses:**

1. The complexity of the coarse-to-fine pipeline is increased by multiple tool calls. The authors do not provide the average number of turns per sample, which is a critical missing metric for a complete evaluation of the true inference efficiency.

2. The Box Adjustment Reward uses a hard, dataset-derived threshold ($\gamma$) to prioritize small crops. This rigid prior risks confining the model's exploratory behavior and severely limiting its generalization ability on scenarios where a large, global context is genuinely required.

**Questions:**

Q1. Why is the Performance Gain So Small on HR-Bench 4K? According to Table 2, under the 1280×28×28 constraint, ERGO's gain on HR-Bench 4K is much lower than its gain on HR-Bench 8K and V*. Could the authors provide a fundamental analysis to explain why ERGO's advantage is significantly suppressed in the HR-Bench 4K scenario?

Q2. Performance and Efficiency Comparison without Resolution Constraint. If the initial input resolution were relaxed to be close to or equal to the original resolution, how would ERGO's performance and efficiency be affected?

Q3. Testing Generalization on Complex High-Resolution Tasks. The high-resolution benchmarks evaluated in this paper, such as V* and HR-Bench, are relatively simple. Given that the coarse-to-fine strategy might be more effective in simpler scenarios, could the authors provide performance data for ERGO, including a comparison against relevant works (DeepEyes, PixelReasoner, TreeVGR, Mini-O3), on more complex benchmarks that require multi-step reasoning, logical judgment, and finer targets (e.g., Visual-Probe[1] or TreeBench[2])?

[1] Mini-o3: Scaling Up Reasoning Patterns and Interaction Turns for Visual Search

[2] Traceable Evidence Enhanced Visual Grounded Reasoning: Evaluation and Methodology

Q4. Quantifying the Multi-Turn Inference Cost. ERGO's inference process is essentially a multi-turn tool calling process. Could the authors compile and report the average number of tool calls per sample on the high-resolution benchmarks, along with a comparison to relevant works, to quantify the overhead of the decision-making process?

---

> ### Author Response · Authors · 2025-11-18
> **Author Rebuttal to uFQT's Comments (1/N)**
>
> > W1 & Q4. Question about the average number of tool-call turns per sample
> - As described in the introductory part of Section 3 (Method), our model uses only **a single tool call** for efficiency-oriented inference. The pipeline is simply: original image + question → bbox → cropped image + question → answer. In contrast, other models such as Deepeyes and Mini-o3 use multiple turns for cropping or for invoking additional tools. As shown in the table below and the revised paper (Table 4), our method **not only delivers higher efficiency but also outperforms these multi-tool-call approaches**.
>
>   | Pixel Const.     | Model      | Max. tool cnt | V*    | Latency (s) |
>   |------------------|------------|---------------|-------|-------------|
>   | 16384×28×28       | Qwen2.5-VL-7B | –             | 77.0  | 4.89        |
>   | **640×28×28**     | DeepEyes | 4         | 64.9 | 3.42 |
>   |                  | DeepEyes   | 2             | 63.9  | 3.07        |
>   |                  | DeepEyes   | 1             | 64.4  | 2.18        |
>   | **640×28×28**     | MiniO3 | 4         | 74.9 | 5.35 |
>   |                  | MiniO3     | 2             | 61.8  | 3.87        |
>   |                  | MiniO3     | 1             | 41.4  | 2.03        |
>   | **640×28×28**     | **ERGO**   | **1**         | **81.7** | **1.61** |
>
>
> - Specifically, DeepEyes uses usually 1 or 2 cropping calls at test time (despite being trained with multiple tool-call turns), and Mini-o3 is trained with up to 6 tool-call turns and evaluated with up to 12 turns. When Mini-o3 is restricted to only 2 turns for efficiency, its cropping behavior becomes unstable and its accuracy declines noticeably. We conjecture that these models rely on multiple tool-call turns because their initial cropping attempts are often unreliable and require iterative correction. In contrast, our method reliably identifies and focuses on the task-relevant region in a single tool call, **achieving both efficiency and accuracy**.
>
>
> ---
>
> > W2. Analysis of box-adjustment reward design
> - We conducted the experiment whether the model prediction is biased by the γ. **Figure 6** in our paper shows that γ = 0.6 generalizes well across multiple unseen datasets with diverse characteristics(**e.g. objects often occupy the full frame**), and that the box-adjustment constant **does not bias ERGO toward fixed-size predictions**.
>   > However, Fig. 6 shows that ERGO infers regions with flexible areas that reflect the underlying characteristics of the data: in MMVP (Tong et al., 2024b), objects often occupy the full frame, whereas in MME-RWL (Zhang et al., 2024), objects are relatively small. (quote from #L364)
>
> - **In the manuscript (Table 6(e)) and the below table**, we report an ablation study on the box-adjustment constant γ ∈ {0.4, 0.6, 0.8}, showing that γ = 0.6 yields the best performance. A larger γ fails to sufficiently penalize overly broad crops, while a smaller γ overly restricts the model, limiting its ability to explore and identify the most informative visual regions.
>
> - We conducted additional experiments exploring alternative designs for the box-adjustment reward in Eq. 2, ranging to an annealed linear form. The annealing function degrades performance. The policy becomes overly focused on shrinking predicted regions, which is easier than improving the region-verification reward. As a result, the model trained with linear annealing fails to reason over semantically meaningful areas. Summarized in the table below and the revised manuscript (Table 6(e), Figure 8, Figure 9), show that the step function remains a reasonable choice.
>
>
>   | γ   | Function        | Average |
>   |-----|------------------|--------:|
>   | 0.4 | step             | 57.8    |
>   | 0.8 | step             | 56.2    |
>   | **0.6** | step             | **58.4** |
>   | 0.6 | linear anneal.   | 51.0    |
>
> ---
>
> > Q1. Clarification on performance gain on HR-Bench 4K (Table 2)
> - The performance gain of our model on HR-Bench-4K is indeed meaningful. On HR-Bench-4K, the original model achieves *71.1% accuracy at 16384x28x28 constraint*. Under the 1280×28×28 constraint, the baselines (PixelReasoner, DeepEyes, TreeVGR, MGPO, VisionThink, Mini-o3) report *66–70% accuracy* (which is less than zero-shot baseline). In contrast, **our model ERGO attains 73.0% accuracy with higher efficiency than the original model**, outperforming all competitive methods.
>
> - We also note that the images in HR Bench-4K are cropped subsets of those in HR Bench-8K, while the question–answer pairs remain identical. Under the same pixel constraints, the target object undergoes substantially stronger degradation in HR Bench-8K, highlighting that our model can effectively reason even when the target object becomes more discernible.
> Also, V* Bench is designed for visual search and therefore contains relatively small target objects, whereas HR Bench empirically features larger objects. This difference explains the performance gain observed across the two benchmarks.

---

> ### Author Response · Authors · 2025-11-18
> **Author Rebuttal to uFQT's Comments (2/N)**
>
> > Q2. Analysis of performance and efficiency without resolution constraints
> - Thank you for your comment. We evaluated the performance and efficiency of ERGO without any resolution constraints, as shown in **the table below**. The results indicate that ERGO maintains strong performance even under highly efficient settings that use a small number of image pixels, while accuracy continues to improve when more pixels are used at the cost of reduced efficiency.
>   |Model| Pixel Const. | V* Bench   | latency (s) |
>   |-------|-------|------|--------------|
>   |Qwen2.5-VL-7B-Inst.| 16384x28x28  | 77.0 | 4.89 |
>   |ERGO| 480x28x28  | 72.8 | 1.06         |
>   || 640x28x28   | 81.7 | 1.61         |
>   || 1280x28x28  | 83.8 | 2.58         |
>   || 1920x28x28  | 83.8 | 3.98         |
>   || 3840x28x28  | 84.8 | 7.20         |
>   || 16384x28x28 | 85.3 | 8.35         |
>
> ---
>
> > Q3. Performance comparison on challenging high-resolution benchmarks
> - We appreciate your constructive suggestion. In response, we conducted additional evaluations comparing ERGO with the relevant works you mentioned on more challenging benchmarks, including _Visual-Probe and TreeBench_. As shown in **the results below and the revised paper (Table 2)**, ERGO outperforms recent methods such as DeepEyes, PixelReasoner, TreeVGR, and _Mini-o3_. ERGO’s reasoning-driven perception allows it to reliably identify task-relevant regions and deliver accurate answers, whereas many baseline models struggle to perform robust reasoning under efficiency-oriented settings with pixel constraints.
>
>
> | Model        | V*   | HR 4K | HR 8K | MME-RWL | TreeBench | VisualProbe | Average |
> |--------------|------|-------|-------|--------|-----------|-------------|---------|
> | PixelReasoner | 74.5 | 66.9  | 61.3  | 49.8   | 40.4      | 31.8        | **52.1**    |
> | DeepEyes      | 78.5 | 66.0  | 60.0  | 48.9   | 40.2      | 39.0        | **52.5**    |
> | TreeVGR       | 76.4 | 66.4  | 60.4  | 47.5   | 48.2  | 26.6        | **53.1**    |
> | Mini-o3        | 81.2 | 70.0 | 65.9 | 36.7  | 36.3      | 39.0        | **52.2**    |
> | MGPO          | 77.5 | 69.8  | 61.1  | 44.4   | 39.3      | 33.4        | **52.6**    |
> | VisionThink   | 73.8 | 66.1  | 65.8  | 49.0   | 41.0      | 17.3        | **49.4**    |
> | **ERGO**       | 83.8 | 73.0 | 69.9| 52.6 | 41.7 | 42.7 | **58.4** |
> ##### Pixel Const.: 1280×28×28
>
>
> | Model        | V*   | HR 4K | HR 8K | MME-RWL | TreeBench | VisualProbe | Average |
> |--------------|------|-------|-------|--------|-----------|-------------|---------|
> | PixelReasoner | 67.2 | 66.5  | 59.9  | 47.7   | 42.3      | 23.9        | **48.9**    |
> | DeepEyes      | 64.9 | 64.4  | 58.3  | 48.9   | 38.5      | 26.8        | **48.1**    |
> | TreeVGR       | 67.0 | 62.4  | 54.4  | 47.5   | 49.1  | 24.1        | **49.2**    |
> | MiniO3        | 74.9 | 62.8  | 57.3  | 34.4   | 36.0      | 31.7        | **48.2**    |
> | MGPO          | 67.5 | 62.8  | 57.3  | 44.4   | 42.0      | 29.7        | **48.7**    |
> | VisionThink   | 61.8 | 66.9  | 60.1  | 46.6   | 39.5      | 17.9        | **45.9**    |
> | **ERGO**       | 81.7 | 67.1 | 66.1 | 49.6 | 43.2 | 35.0 | **55.2** |
> ##### Pixel Const.: 640×28×28

---

### Official Review · Reviewer_6jTD · 2025-10-31

**Soundness:** 2
**Presentation:** 3
**Contribution:** 3
**Rating:** 4
**Confidence:** 3

**Summary:**

The authors introduce ERGO (Efficient Reasoning & Guided Observation), a novel rl-based framework for efficient high-resolution visual question answering in VLMs. The key innovation lies in a reasoning-driven perception pipeline that identifies semantically relevant regions using low-resolution inputs and re-encodes those regions at full resolution for fine-grained reasoning. Unlike traditional perception-driven approaches, ERGO explicitly integrates multimodal reasoning into region selection through a two-stage coarse-to-fine pipeline and a task-driven contextual exploration reward in the RL setup. Experiments across multiple high-resolution VQA benchmarks demonstrate strong performance and efficiency, outperforming state-of-the-art models in both accuracy and computational cost.

**Strengths:**

1. Introduces a principled RL framework that aligns vision token efficiency with VQA performance using a custom TCE reward.
2. Paper is well-written, figures are illustrative, and reward function formulations are clear .

**Weaknesses:**

1. While the RL approach is justified, comparison with alternative non-RL cropping or saliency-based models is missing.
2. Qwen2.5 is used as both the base and reward model. It's unclear whether the approach generalizes well to other vision backbones.
3. The paper assumes that an RL formulation is the optimal paradigm for guiding region selection, yet no strong justification is provided for why this should outperform supervised learning approaches (e.g., using ground truth regions, attention supervision, or self-training).
4. The proposed reward structure (especially the region-verification reward) seems to mimic supervised alignment between image regions and answers, raising the question of whether RL introduces unnecessary complexity.
5. The combination of region-verification and box-adjustment rewards appears empirically effective, but the design choices (e.g., fixed $\gamma$ = 0.6, hard step function in Eq. 2) are purely heuristic.
6. The core contribution is the “reasoning-driven perception” strategy, yet the evaluation focuses heavily on end-to-end VQA accuracy, without thoroughly validating whether the selected regions are actually semantically aligned with the question (e.g, via IOU with GT regions or attention overlap with human fixations). Figure 5 attempts to address this, but the metric is poorly defined, lacks baseline comparisons, and only handles binary masking scenarios.
7. The reward model is frozen, but still based on Qwen2.5-VL-72B, which is the same family as the base policy model (Qwen2.5-VL-7B). This introduces a risk of reward leakage, particularly if the reward model and policy share weights or similar training objectives.

**Questions:**

See Weaknesses Above.

---

> ### Author Response · Authors · 2025-11-18
> **Author Rebuttal to 6jTD’s Comments (1/N)**
>
> >W1. Comparison with additional baselines
>
> - We appreciate the reviewer's suggestion and have expanded our baseline comparisons to include the Visual-CoT, Chain-of-Spot models and zero-shot baselines. As shown in the results below and the updated manuscript (Table 2), our RL-only model **surpasses** the other RL-only, SFT-only and SFT-then-RL.
>
>  | Model   | Training Method | V* Bench | HR Bench 4K | HR Bench 8K | MME-RW Lite | TreeBench | VisualProbe | Average |
>  |------------------|------------------------------|---------:|------------:|------------:|-------------:|----------:|------------:|--------:|
>  | Qwen2.5-VL-7B-Inst. | zero shot |  56.5  |65.6 |  56.5 |  42.6  |  40.0  |  15.2  |   **44.8**  |
>  | Vis-CoT | only SFT |  56.5  | 46.6 |  40.8 |  35.6  |   30.6  |  23.5  |   **38.0**  |
>  | Chain-of-Spot | only SFT |  48.4  | 41.1 |  36.5 | 31.6  | 32.1  | 18.1  |   **33.8**  |
>  | PixelReasoner | SFT -> RL |    74.5  |        66.9 |        61.3 |        49.8  |     40.4  |       31.8  |   **52.1**  |
>  | TreeVGR    | SFT -> RL |  76.4  |        66.4 |        60.4 |        47.5  |   48.2 |       26.6  |   **53.1**  |
>  |Mini-o3     | SFT -> RL |  81.2 |       70.0 |        65.9 |        36.7  |     36.3  |       39.0  |   **52.2**  |
>  |MGPO      | only RL |  77.5  |        69.8 |        61.1 |        44.4  |     39.3  |       33.4  |   **52.6**  |
>  | DeepEyes   | only RL |  78.5  |        66.0 |        60.0 |        48.9  |     40.2  |       39.0  |   **52.5**  |
>  | **ERGO**  | only RL  | 83.8 |     73.0 |     69.9 |     52.6 |   41.7 |   42.7 | **58.4** |
> ---
>
> >W2 & W7. Analysis of different reward models' impact
>
> - We thank the reviewer for this insightful comment. Our initial experiments used Qwen2.5-VL-72B-Instruct as the reward model. In response to the suggestion, we also evaluated ERGO with other reward model types, specifically InternVL3-78B and GLM4.5V-108B, while keeping all other training configurations the same. The results, provided below and the updated manuscript (Table 6(d)), show that **our reward formulation generalizes well across different reward model types**.
>
>    | Reward model       | Average |
>    |-------------------|---------|
>    | InternVL3-78B      | 57.2    |
>    | GLM4.5V-108B       | 58.2    |
>    | Qwen2.5-VL-72B     | **58.4**    |
>
> ---
>
> >W3. Justification for choosing RL over supervised alternatives for region selection.
>
> - Thank you for the comment. As demonstrated in our W1 experiments, our RL-only approach clearly surpasses the SFT-only, SFT-then-RL baselines.
>
> - Moreover, the RL-only framework significantly reduces data-construction overhead. RL training relies solely on question-answer (Q&A) pairs, whereas SFT requires more elaborate supervision such as **bounding boxes and reasoning-path annotations [1,2,3,4].** By removing the need for these **labor-intensive labels**, our method achieves much better scalability.
>
> - Attention-based supervision is also challenging to adopt in modern LLM/VLM training and inference pipelines. In practice, extracting or modifying attention scores is non-trivial due to the **widespread use of highly optimized Flash Attention kernels [5]**, which limit direct access to attention weights.
>
> - Finally, our GRPO-based RL training can be viewed as a form of self-training: it requires **no intermediate annotations (e.g., bounding boxes or text reasoning paths)**, yet the model automatically learns where to focus and how to produce effective reasoning trajectories.
>
>   ###### [1] Wang, Haochen, et al. "Traceable evidence enhanced visual grounded reasoning: Evaluation and methodology." arXiv preprint arXiv:2507.07999 (2025).
>
>   ###### [2] Shao, Hao, et al. "Visual cot: Advancing multi-modal language models with a comprehensive dataset and benchmark for chain-of-thought reasoning." Advances in Neural Information Processing Systems 37 (2024): 8612-8642.
>
>   ###### [3] Lai, Xin, et al. "Mini-o3: Scaling up reasoning patterns and interaction turns for visual search." arXiv preprint arXiv:2509.07969 (2025).
>
>   ###### [4] Su, Alex, et al. "Pixel reasoner: Incentivizing pixel-space reasoning with curiosity-driven reinforcement learning." arXiv preprint arXiv:2505.15966 (2025).
>
>   ###### [5] Wen, Zichen, et al. "Token Pruning in Multimodal Large Language Models: Are We Solving the Right Problem?." arXiv preprint arXiv:2502.11501 (2025).
>
> ---

---

> ### Author Response · Authors · 2025-11-18
> **Author Rebuttal to 6jTD’s Comments (2/N)**
>
> >W4. Justification for the region-verification reward
> - As noted in our responses to W1 and W3, RL training achieves strong accuracy while significantly reducing data-construction effort.
>
> - Supervised alignment between image regions and answers **requires detailed annotations indicating which regions correspond to each Q&A pair, often together with text reasoning trajectories**. Although automatic pipelines can help generate such labels, high-quality data typically still requires human curation. RL avoids this overhead by relying only on Q&A pairs, making it **more annotation-efficient and scalable**.
>
> - RL provides better generalization by allowing the model to explore diverse solution paths and learn the underlying decision process [1,2,3]. Overall, our reward formulation, including the region-verification reward, effectively guides the model to focus on the correct regions in high-resolution images to produce accurate answers.
>
>   ###### [1] Chen, Liang, et al. R1-V: Reinforcing Super Generalization Ability in Vision-Language Models with Less Than $3. 2025.
>   ###### [2] Zhu, Hanqing, et al. "The Path Not Taken: RLVR Provably Learns Off the Principals." arXiv preprint arXiv:2511.08567 (2025).
>   ###### [3] Shenfeld, Idan, Jyothish Pari, and Pulkit Agrawal. "RL's Razor: Why Online Reinforcement Learning Forgets Less." arXiv preprint arXiv:2509.04259 (2025).
>
> ---
>
> >W5. Analysis of box-adjustment reward design choices and objective function weightings
>
> - In the manuscript (Table 6(e)), we present an ablation on the box-adjustment constant γ ∈ {0.4, 0.6, 0.8}, showing that γ = 0.6 provides the best balance: larger values insufficiently penalize overly broad crops, while smaller ones overly constrain the model's ability to explore informative regions. Figure 6 in the manuscript further demonstrates that the model generalizes well across unseen datasets with diverse characteristics and does not bias ERGO toward fixed-size predictions.
>
>   | γ   | Function        | Average |
>   |:---:|:---------------|:-------:|
>   | 0.4 | step           | 57.8    |
>   | 0.8 | step           | 56.2    |
>   | **0.6** | step       | **58.4** |
>
>
>
> - Based on the reviewer's comment, we conducted additional experiments exploring alternative designs for the box-adjustment reward in Eq. 2, ranging from a hard step function to an annealed linear form. The annealing reward function degrades performance. The policy becomes **overly focused on shrinking predicted regions**, which is easier than improving the region-verification reward. As a result, the model trained with linear annealing fails to reason over semantically meaningful areas. Summarized in the table below and the revised manuscript (Table 6(e), **Figure 8, Figure 9**), show that the step function remains a reasonable and effective design choice.
>
>   | γ   | Function        | Average |
>   |:---:|:---------------|:-------:|
>   | **0.6** | step       | **58.4** |
>   | 0.6 | linear anneal. | 51.0    |
>
>
> - We further investigate multiple configurations of $\alpha$ and $\beta$. Configuration (ii), where the box-adjustment weight $\beta$ is larger than $\alpha$, shows the lowest performance. In contrast, configuration (iii), which scales both $\alpha$ and $\beta$ by a common multiplicative factor **while preserving their relative ratio, outperforms configuration (i)** even when the region verification signal is weakened. This further **underscores the strength of our TCE reward** formulation and validates the proposed weighting strategy.
>
>   | No.  | (α, β)       | Avg.  |
>   |:----:|:------------:|:-----:|
>   | (i)  | (1.0, 1.0)   | 56.2  |
>   | (ii) | (1.0, 2.0)   | 54.3  |
>   | (iii)| (0.5, 0.25)  | 56.8  |
>   | (iv) | **(1.0, 0.5)** | **58.4** |
>
> ---

---

> ### Author Response · Authors · 2025-11-18
> **Author Rebuttal to 6jTD’s Comments (3/N)**
>
> >W6. Validation of semantic alignment between selected regions and questions
>
> - Thanks for your comment. To assess the semantic alignment between the selected regions and the questions, we further evaluate the success ratio of the predicted regions. Specifically, we measure the fraction of each target object covered by ERGO's detected bounding boxes on the V* dataset. The table below report these results, showing that the predicted boxes consistently capture the target objects.
>
>
>
>   | Setting                   | Metric                 | ERGO   | DeepEyes | MGPO  | TreeVGR |
>   |---------------------------|------------------------|--------|----------|-------|---------|
>   | naive                     | Target Coverage Score | **80.4** | 59.1     | 57.9  | 63.8    |
>   |                           | Success Ratio@0.5     | **80.9** | 64.7     | 56.5  | 69.9    |
>   | target obj. masked        | Target Coverage Score | **57.9** | 29.2     | 19.2  | 19.3    |
>   |                           | Success Ratio@0.5     | **57.3** | 28.3     | 17.5  | 17.0    |
>   | naive - target obj. masked     | Target Coverage Score | **22.5** | 29.9     | 38.7  | 44.5    |
>   |                           | Success Ratio@0.5     | **23.6** | 36.4     | 39.0  | 52.9    |
>
>
>    ###### ** The proportion target object bounding box (bbox) included by the predicted bbox has been labeled as success.
>
>    ###### ** If the number of target object is larger than 1, then we used average of success rate (e.g. if 1 of 2 target obj. is included, the score is 0.5)
>
> - Furthermore, we clarify the analysis in **Figure 5 and above table**, which evaluates whether the predicted regions accurately cover the ground-truth target objects by computing the intersection between the predicted and GT bounding boxes. This evaluation considers both the masked and unmasked settings, including comparisons with multiple baselines.
>
>   - Target-object masked setting
> 	- The target object is fully masked in black, preventing the model from using any optical appearance cues. In this case, the only information the model can rely on is non-visual (semantic/ contextual) reasoning to locate the target.
>   - Naive setting
> 	- The model can still potentially exploit the optical appearance of the target object, even if it is very small.
>
>   By comparing performance under these two settings, we can quantify how much the model is capable of locating the target semantically rather than visually. We acknowledged the needs for clearer representation of our analysis, thus we updated our equation for clarity in updated manuscript #L359.
>
>   | Setting                   | Metric                 | ERGO   | DeepEyes | MGPO  | TreeVGR |
>   |---------------------------|------------------------|--------|----------|-------|---------|
>   | naive - target obj. masked| Target Coverage Score | **22.5** | 29.9     | 38.7  | 44.5    |
>   |                           | Success Ratio@0.5     | **23.6** | 36.4     | 39.0  | 52.9    |
>   ###### ** Less value means more robustness to optical appearance.

---

> ### Comment · Reviewer_6jTD · 2025-11-25
>
> Thanks to the authors for the detailed response. This has addressed most of my concerns.

---

> > ### Author Response · Authors · 2025-11-27
> > **Response to Reviewer 6jTD**
> >
> > Thanks for your thoughtful consideration. We appreciate the time and effort you have dedicated to reviewing our work. Please let us know if any additional clarification would be helpful for better work.

---

### Official Review · Reviewer_zSHE · 2025-10-31

**Soundness:** 3
**Presentation:** 3
**Contribution:** 2
**Rating:** 6
**Confidence:** 4

**Summary:**

This paper presents ERGO, a model designed to tackle high-resolution visual understanding for vision-language models without burning through too much computing power. The core idea is a two-stage pipeline: first, it analyzes a low-res version of the image to spot task-relevant regions, then crops only those areas at full resolution for detailed reasoning. Unlike other methods that struggle when objects get blurry after downsampling, ERGO uses reasoning-driven perception to find important regions. Trained with a reinforcement learning framework that includes rewards for good region selection, efficient crop size, and accurate answers, ERGO delivers impressive results. It outperforms models like Qwen2.5-VL-7B on various benchmarks. Even with strict pixel constraints, it maintains high accuracy across multiple high-res and conventional multimodal benchmarks.

**Strengths:**

1. The reasoning-driven perception approach is super smart. By using context to find relevant regions instead of relying on clear visual cues, it handles blurry low-res inputs way better than existing perception-driven models.

2. The authors propose a well-designed reward system in the RL framework, combining rewards for region quality, crop size, and answer accuracy, makes sure the model learns to be both effective and efficient.

3. The results are really practical. The model delivers real latency improvements and doesn’t lose out on performance when applied to other multimodal tasks beyond high-res reasoning.

**Weaknesses:**

1. The author mentions using a frozen reward model to provide the region reward. Analysis of how the reward model performs is crucial for others to validate the effectiveness of this reward design.

2. I appreciate the authors' effort in providing the predicted area ratio. However, I also want to know the success ratio of the predicted area for benchmark evaluation. This serves as an important metric to analyze how the model perform and the effevctiveness of model training.

3. The authors should provide more baseline method, like using the SFT method to performance area cropping in previous work (Visual-CoT and Chain-of-Spot), also the zero-shot baseline and tool-calling baseline should be included.

**Questions:**

As stated in weakness part.

---

> ### Author Response · Authors · 2025-11-18
> **Author Rebuttal to zSHE’s Comments**
>
> >W1. Analysis of how the reward model performs and its effectiveness
>
> - We appreciate this constructive comment. We acknowledge that our region verification reward is indeed an approximation (a proxy) for the "optimal, self-contained zoom-in region". The explicit, large-scale annotation of optimal regions is **labor-intensive and subjective**, representing a practical bottleneck. By using a more practical and scalable region-verification reward, we **reframe the task of finding optimal zoom-in region to a relatively easier task** of answering the question with the given single image. This approach allows the model to identify salient regions **without** requiring perfect, fully-annotated zoom-in labels.
>    > This design encourages the policy model to identify informative, task-relevant regions that preserve
> sufficient information for accurate reasoning, without the need for additional annotations.
>
>    We mentioned the reward mechanism at the submitted manuscript like above, but we further emphasized our acknowledgement at #L177 in the revised manuscript.
>
> - The following table and the updated results (Table 6(c) & (d)) validate that this provides a highly effective solution. These findings indicate that our reward design is generalizable and relatively easy proximation across different reward model sizes and types.
>
>    | Reward model       | Average |
>    |-------------------|---------|
>    | Qwen2.5-VL-3B             | 55.6    |
>    | Qwen2.5-VL-7B             | 56.4    |
>    | InternVL3-78B      | 57.2    |
>    | GLM4.5V-108B       | 58.2    |
>    | Qwen2.5-VL-72B     | **58.4**    |
>
>    ##### TreeVGR, best scoring on average other than ours, scores 53.1
>
> ---
> > W2. Analysis of the success ratio for predicted areas
> - Thanks to your comment, we analyze the success ratio of the predicted areas as an additional metric to assess model performance and the effectiveness of training. We compute the proportion of the target object contained inside the detected bounding boxes of ERGO on the V* benchmark dataset. The table below shows that predicted boxes reliably capture the target objects. We also notify that **the success ratio closely aligns with the target coverage score without masking**, which evaluates whether the predicted region covers the ground-truth object.
>
>    >To quantify this ability, we analyze whether the predicted region covers the GT target object, by defining Target Coverage Score (quote from submitted manuscript #L349)
>
>    We acknowledged the needs for clearer representation of our analysis, thus we updated our equation for clarity (#L358 in the revised manuscript).
>
>    | Setting          | ERGO  | DeepEyes | MGPO  | TreeVGR |
>    |-----------------|-------|----------|-------|---------|
>    | Target Coverage Score       | **80.4**  | 59.1     | 57.9  | 63.8  |
>    | Success ratio for predicted area         | **80.9** | 64.7    | 56.5 | 69.9   |
>
>    ###### ** The proportion target object bounding box (bbox) included by the predicted bbox has been labeled as success.
>
>    ###### ** If the number of target object is larger than 1, then we used average of success rate (e.g. if one of two target object is included, the score is 0.5)
> ---
> > W3. Comparison with additional baselines (SFT, zero-shot, tool-calling)
> - We agree with your suggestion, and we extended our baselines to include Visual-CoT, Chain-of-Spot, and Mini-o3 models for comparison with ERGO. The results below and the revised paper (Table 2) show that our RL-only model outperforms the supervised finetuning (SFT)-only, SFT-then-RL and tool-calling.
>
>
>  | Model   | Training Method | V* Bench | HR Bench 4K | HR Bench 8K | MME-RW Lite | TreeBench | VisualProbe | Average |
>  |------------------|------------------------------|---------:|------------:|------------:|-------------:|----------:|------------:|--------:|
>  | Qwen2.5-VL-7B-Inst. | zero shot |  56.5  |65.6 |  56.5 |  42.6  |  40.0  |  15.2  |   **44.8**  |
>  | Vis-CoT | only SFT |  56.5  | 46.6 |  40.8 |  35.6  |   30.6  |  23.5  |   **38.0**  |
>  | Chain-of-Spot | only SFT |  48.4  | 41.1 |  36.5 | 31.6  | 32.1  | 18.1  |   **33.8**  |
>  | PixelReasoner | SFT -> RL |    74.5  |        66.9 |        61.3 |        49.8  |     40.4  |       31.8  |   **52.1**  |
>  | TreeVGR    | SFT -> RL |  76.4  |        66.4 |        60.4 |        47.5  |   48.2 |       26.6  |   **53.1**  |
>  |Mini-o3     | SFT -> RL |  81.2 |       70.0 |        65.9 |        36.7  |     36.3  |       39.0  |   **52.2**  |
>  |MGPO      | only RL |  77.5  |        69.8 |        61.1 |        44.4  |     39.3  |       33.4  |   **52.6**  |
>  | DeepEyes   | only RL |  78.5  |        66.0 |        60.0 |        48.9  |     40.2  |       39.0  |   **52.5**  |
>  | **ERGO**  | only RL  | 83.8 |     73.0 |     69.9 |     52.6 |   41.7 |   42.7 | **58.4** |

---

> > ### Comment · Reviewer_zSHE · 2025-11-25
> >
> > The author's rebuttal addresses my concerns and seems promising. Therefore, I will maintain my positive score.

---

> > > ### Author Response · Authors · 2025-11-27
> > > **Response to Reviewer zSHE**
> > >
> > > Thanks for your follow-up response. We appreciate the time and effort you have dedicated to reviewing our work. Please let us know if any additional clarification would be helpful for better work.

---

### Author Response · Authors · 2025-11-18
**Rebuttal by Authors**

We sincerely appreciate the reviewers’ thoughtful feedback. We have carefully addressed each comment and revised the paper accordingly (with changes marked in blue). We would be happy to engage in any further productive discussions.

---
### Strengths Noted by Reviewers

- We are pleased that the reviewers recognize the importance of our study in enabling efficient high-resolution image processing for LVLMs. They highlighted that our reasoning-driven perception mechanism and two-stage coarse-to-fine pipeline provide a _super-smart_ and _effective_ solution to the limitations of existing perception-driven reasoning approaches [zSHE, 9PNo]. Reviewers also noted our _well-motivated_ and _principled_ RL framework with its novel reward formulations [zSHE, 6jTD, 9PNo]. They further acknowledged the _practicality_ and _strong performance_ of our method, which achieves efficiency gains by processing fewer image tokens while still outperforming recent competitive baselines [zSHE, uFQT, 9PNo]. The _clarity_ of our analyses, writing, and presentation was also commended [zSHE, 6jTD].

---
### Additional Experiments per Reviewers’ Suggestions

We have conducted numerous additional experiments to thoroughly address the reviewers’ comments:
- Training with different reward model backbones
- Training with different reward weighting factors
- Training with different design choices of the box-adjustment reward
- Comparisons with various baselines, including SFT-based, tool-calling methods
- Evaluation on additional datasets such as TreeBench and Visual Probe
- Evaluation without a fixed pixel budget, including comparisons with baselines under their original configurations and under latency-constrained settings.
- Analysis of the impact of the box-adjustment reward function on training
- Analysis of performance and latency across a wide range of resolutions
- Analysis on trade off between latencies and tool calling counts
- Analysis of the success ratio, i.e., the proportion of the target object inside predicted boxes

---

### Author Response · Authors · 2025-11-24
**A Gentle Reminder for the Author-Reviewer Discussion**

Dear Reviewers,

Thank you for the time and effort you have dedicated to evaluating our paper. We would be grateful if you could share your thoughts on our rebuttal. We have carefully addressed all of your comments and are ready to engage in constructive discussion.

Sincerely,
Authors

---

### Author Response · Authors · 2025-12-03
**Summary of Rebuttal phase**

Dear AC,

We sincerely thank all reviewers for their insightful and constructive feedback. In response, we have revised the manuscript, provided detailed responses to each comment, and addressed concerns raised during the review process.

During the active discussion period, we conducted new experiments and added detailed clarifications. All new results matched expectations and directly addressed the reviewers’ comments. We also received **positive feedback from Reviewers zSHE and 6jTD**, who noted that our responses were promising and had resolved their concerns.

For the remaining points, further discussion unfortunately could not proceed due to the OpenReview incident. **Reviewer uFQT** requested results without resolution constraints, evaluation with tool-call turns per sample, and further explanation of the box-adjustment reward. During discussion, **Reviewer 9PNo** raised concerns regarding fixed-constraint evaluation; we clarified the evaluation setup and provided additional experiments comparing baselines under their original configurations and under latency constraints. We prepared detailed follow-up responses and accompanying experiments for each of these points, but the incident prevented further discussion.

Finally, we highlight that, to the best of our knowledge, our work is the first in visual–language reasoning to demonstrate practical latency reduction without sacrificing performance, and in fact with performance improvements.

Thank you again for your time and consideration.

Best regards,
Authors

---

### Meta-Review · Area_Chair_xi7F · 2026-01-07

**Summary:**

This paper proposes an efficient-RL based method for VQA of high resolution images. The key contribution of this work is to propose an RL-based approach to finding the ROI from a low resolution input and then creating a high-resolution crop from it for subsequent VQA. All the design choices of the proposed approach are well motivated by intuition or experimentation. Via extensive experimentation the authors show that their approach significantly outperforms many existing baselines in terms of accuracy at low resolution inputs and also at higher resolution inputs in terms of latency.

Four reviewers provided scores of (4, 6, 6, 4). Most of the reviewers' concerns were adequately addressed by the authors rebuttal and all reviewers were positive towards recommending acceptance. The AC concurs with the reviewers' consensus and recommends acceptance. The authors should include the additional results that they presented in the rebuttal into the final version of their paper.

Congratulations!

**Reviewer Concerns:**

Concerns addressed:
* analysis of the reward model
* additional ablations
* additional comparisons to baselines
* accuracy of bounding box prediction
* justification of many design choices and hyper parameters

**Reviewer Scores:**

1. Reviewer zSHE (Rating: 6: marginally above the acceptance threshold. But would not mind if paper is rejected)

The reviewer's primary concerns were: (a) analysis of how the reward model performs, (b) presenting the accuracy of bounding box prediction and (c) showcasing additional baselines results. The reviewer mentioned "The author's rebuttal addresses my concerns and seems promising. Therefore, I will maintain my positive score."

2. Reviewer 6jTD (Rating: 4: marginally below the acceptance threshold. But would not mind if paper is accepted)

This reviewer's primary concerns were (a) additional comparisons to other approaches, (b) generalization to other vision backbones, (c) justification for why RL is best for region selection and for the hyper parameters chosen, and (d) an analysis of the selected ROI. The reviewer mentions "Thanks to the authors for the detailed response. This has addressed most of my concerns."

3. Reviewer uFQT (Rating: 6: marginally above the acceptance threshold. But would not mind if paper is rejected)

This reviewer's primary concerns were (a) that the complexity of the coarse-to-fine pipeline is increased by multiple tool calls and (b) the Box Adjustment Reward uses a hard threshold, (c) questions about the performance on HR-Bench-4K, (d) performance evaluation of ERGO at a higher resolution and (e) performance evaluation on additional high-resolution benchmarks. The authors provided detailed answers and additional experiments that addressed each of these concerns. The reviewer is likely to have maintained their positive rating.

4. Reviewer 9PNo (Rating: 4: marginally below the acceptance threshold. But would not mind if paper is accepted)

The reviewer's primary concerns were (a) that it is a complex system because of multiple reward models, (b) needs an ablation study on the weights of the rewards and (c) required an evaluation of the results of baseline models at their original settings of resolution for which they were designed. The reviewer mentioned that "I intend to maintain my original score, but would not mind if the paper is accepted." The authors provided numerical comparisons of all methods at higher resolutions during the rebuttal and even in this setting their method performs comparably to the baselines in terms of accuracy while achieving higher speed.

---

### Decision · Program_Chairs · 2026-01-26

Accept (Poster)